# DODA: DIFFUSION FOR OBJECT-DETECTION DOMAIN ADAPTATION IN AGRICULTURE

## ABSTRACT

Object detection has wide applications in agriculture, but the trained models often struggle to generalize across diverse agricultural environments. To address this challenge, we propose DODA (Diffusion for Object-detection Domain Adaptation in Agriculture), a unified framework that leverages diffusion models to generate domain-specific detection data for multiple agricultural scenarios. DODA incorporates external domain embeddings and an improved layout-to-image (L2I) approach, allowing it to generate high-quality detection data for new domains without additional training. We demonstrate DODA's effectiveness on the Global Wheat Head Detection dataset, where fine-tuning detectors on DODA-generated data yields significant improvements across multiple domains (maximum +15.6 AP). DODA provides a simple yet powerful approach to adapt object detectors to diverse agricultural scenarios, lowering barriers for more plant breeders growers to use detection in their personalized environments.

## 1 INTRODUCTION

Object detection has been widely used in various aspects of agriculture, such as yield estimation (Wang et al., 2022b;c), disease identification (Wu et al., 2021; Zhang et al., 2020), and decision-making support (Bazame et al., 2021; Wang et al., 2023b). These applications can improve the efficiency and profitability of plant breeders and farmers to improve their efficiency and profits. However, these models are built for specific settings, farms, crop varieties and management systems and excel primarily in their specific settings. However, agricultural scenarios are very diverse, and domain shifts caused by factors such as crop varieties, growth stages, cultivation management, and imaging pipeline, making directly applying a given model to new environments unfeasible.

The diversity of agricultural scenes makes domain adaptation (DA) a key focus in this field. DA can be divided into instance-level DA and image-level DA (Ma et al., 2022) (see related work for more details). Instance-level DA (Ma et al., 2021) typically involves training a discriminator to separate domain-specific features, aligning the features across source and target domains. On the other hand, image-level DA (Gogoll et al., 2020; Zhang et al., 2021) aligns the image styles of the two domains through GAN (Goodfellow et al., 2014) or Fourier transform (Yang & Soatto, 2020), but the general shape and distribution of objects (both foreground and background) in the transformed image are basically fixed. To achieve optimal results, additional instance-level DA is often required to address the remaining differences (Ma et al., 2022). For new scenes, instance-level DA and GAN-based methods require retraining the entire network, which limits scalability and substantial technical barriers remain for non-experts.

Recently, diffusion models (Ho et al., 2020; Rombach et al., 2022) have attracted attention for their ability to generate high-fidelity and novel images that do not appear in the training set. A growing number of studies explore the potential for diffusion models to address data-related challenges, including visual representation learning (Tian et al., 2024; 2023), classification (Azizi et al., 2023; Sarıyıldız et al., 2023), and semantic segmentation (Schnell et al., 2023; Tan et al., 2023; Xie et al., 2023). Existing diffusion-based methods to generate detection data can be categorized into three types. 1. Copy-paste Synthesis (Ge et al., 2022; Lin et al., 2023): foreground and background are synthesized separately (or just the foreground) and then combined, which often results poor image consistency. 2. Direct Image Generation (Zhang et al., 2023b; Feng et al., 2024): images are generated via a text-to-image model, with labels obtained from a detector or module, which

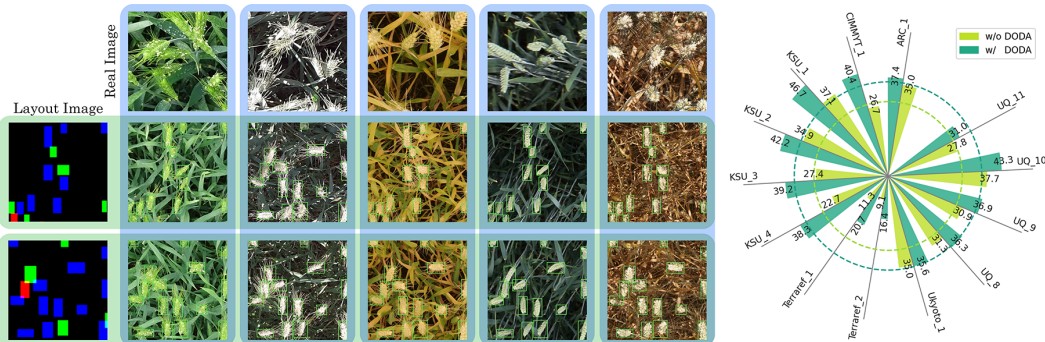

Figure 1: Overview. Left, We propose **DODA** to generate detection data for diverse agricultural domains, the context of the generated images matches the target domain, and the layout of the generated images aligns with the input layout images. Right, fine-tuning detector on DODA-generated data yields significant improvements across multiple domains.

produces consistent images, but isn't suitable for DA because the detector can't operate in the unseen domain without additional detection labels. 3. Layout-to-Image (L2I) Generation (Chen et al., 2023; Zheng et al., 2023; Cheng et al., 2023): semantic layout are used as guidance to control the layout of generated images. L2I relies on a layout encoder to encode the layout, which requires image-label pairs for training. Consequently, the generated data are typically used to enhance its training set and are not suitable for DA. These raise the question: *How can diffusion model be leveraged to generate high-quality detection data for specific domains?*

To address these challenges, we propose DODA, a unified framework for generating high-quality detection data across diverse agricultural domains. By conditioning the diffusion model with external domain embeddings, DODA decouples the learning of domain-specific features from the model, allowing it to generate images for target domains without additional training. Furthermore, we found that existing L2I methods are overly complicated, leading to poor alignment between layout and image features. For these reasons we introduce a new layout-image-to-image (LI2I) technique to simplify the process. In our method, the layouts are directly represented as images and encoded with a simple vision encoder. This approach greatly improves control over the generated image layout, which is crucial for the label accuracy. On the COCO dataset (Lin et al., 2014), our LI2I method (42.5 mAP) achieves a significant performance improvement, outperforming the previous SOTA L2I method (GeoDiffusion (Chen et al., 2023), +14.8 mAP) and nearly matching the performance on real images (45.2 mAP). To further improve the quality of generated detection data, we suggest dividing DODA's training into pre-training and post-training, and pre-training the model on a larger set of unlabeled agricultural images. To support this, we collected an additional 65k agricultural images. Extensive experiments on the Global Wheat Head Detection (GWHD) dataset (David et al., 2021), which is the largest agricultural detection dataset and includes diverse sub-domains, show consistent improvements across multiple domains (as shown in Fig. 1 right), with a maximum AP increase of 15.6. These results highlight DODA's potential to lower the barriers for growers using detection in their personalized environments.

The main contributions of this paper can be summarized as:

- We decouple the learning of domain-specific features from the diffusion model by incorporating external domain informations, enabling the diffusion model to generate images for target domains without requiring additional training.

- We introduce the LI2I method to enhance layout control and improve label accuracy. Results on the COCO dataset show that our LI2I method generates images with highly accurate layouts, significantly outperforming previous L2I methods and closely approximating real images.

- We demonstrate that pre-training with more unlabeled data significantly enhances the quality of generated data. To facilitate this, we collected and cleaned 65k unlabeled images, doubling the size of the GWHD for pre-training.

- The substantial and consistent AP improvements across multiple domains on the GWHD dataset demonstrate that our synthesized domain-specific detection data effectively helps detectors adapt to new domains.

## 2 RELATED WORK

**Detection and counting in agriculture.** The number, size, and appearance of specific plant organs directly influence yield. Accurately and efficiently quantifying these traits benefits breeders selecting superior varieties and growers optimizing management. Manual data collection, however, is slow and labor-intensive. Therefore, automating these tasks has become a major research focus. An effective method is to use neural networks to generate density maps, and count the plants from the density maps (Hobbs et al., 2021a; Osco et al., 2020; Hobbs et al., 2021b). Detection and segmentation can provide additional contour information, enabling the estimation of crop size, health, and maturity. These techniques have gained popularity, with models developed for crops such as wheat (Khaki et al., 2022), maize (Zou et al., 2020), sorghum (Ghosal et al., 2019), and cranberries (Akiva et al., 2020). Many studies rely on images from individual experimental fields. However, genetic differences, environmental variation and differences in image pipeline lead to huge differences in the agricultural images(Ghosal et al., 2019). This limited these models to generalize to new agricultural scenes.

**Global Wheat Head Detection dataset.** The GWHD (David et al., 2021) dataset is one of the largest agricultural detection datasets, specifically focused on close-range wheat head detection. It consists of 47 sub-domains, each with certain differences, such as location, imaging pipeline, collection time, wheat development stages, and wheat varieties. This division allows the development and evaluation of a robust domain adaptation algorithm that perform well under different agricultural environments.

**Domain shift and adaptation.** Domain shift (DS) includes image-level DS (overall differences in factors such as lighting and color, which subtly affect the distribution of features) and instance-level DS (differences in object pose, category, and position) (Ma et al., 2022). In object detection, DS often causes detectors trained on a source domain to perform poorly when applied to a new target domain. To address this problem, few-shot domain adaption (FDA) (Gao et al., 2023; Nakamura et al., 2022) investigates improving model performance on the target domain by using few labeled target images. In contrast, unsupervised domain adaptation (UDA) (Khodabandeh et al., 2019; Li et al., 2021a) aims to improve performance using only unlabeled target images. The focus on this paper is UDA.

**Layout-to-image generation.** The category and position information of all objects in an image is referred to as layout. Layout-to-image is the task of synthesizing images that align with the given layout. There are two ways to represent layout: 1. Bounding box, which defines the position of an object by four vertices. 2. Mask, which defines the shape and position of an object by the semantic mask. Methods based on bounding box (Chen et al., 2023; Rombach et al., 2022; Zheng et al., 2023) rely on a text encoder to encode the input layout. However, this encoding approach often struggles to align the layout with the image features, leading to weaker control over the layout. On the other hand, mask-based methods (Mask-to-Image , M2I) (Zhang et al., 2023a) allow for more precise control over the layout. Despite this advantage, generating masks algorithmically is challenging and lacks flexibility, making M2I less suitable for data generation.

## 3 METHOD

### 3.1 PRELIMINARIES

Song et al. (Song et al., 2020) provided a new perspective on explaining the diffusion model (Ho et al., 2020) from the perspective of stochastic differential equation (SDE) and score-based generative models (Song & Ermon, 2019; Hyvärinen & Dayan, 2005). The forward diffusion process perturbs the data with random noise, described by the following SDE:

$$d\mathbf{x} = f(\mathbf{x}, t)dt + g(t)\mathbf{w} \tag{1}$$

Where the $f : \mathbb{R}^d \to \mathbb{R}^d$ is the drift coefficient of $\mathbf{x}_t$, $g : \mathbb{R} \to \mathbb{R}$ is the diffusion coefficient of $\mathbf{x}_t$, and $\mathbf{w}$ is the standard Brownian motion.

The forward diffusion process gradually transforms the data from the original distribution $p(\mathbf{x}_0)$ into a simple noise distribution $p(\mathbf{x}_T)$, over time $T$. By reversing this process, we can sample

$\mathbf{x}_0 \sim p(\mathbf{x}_0)$ starting from random noise. According to Anderson (1982), this reverse process is given by a reverse-time SDE:

$$d\mathbf{x} = [f(\mathbf{x}, t) - g(t)^2 \nabla_{\mathbf{x}_t} \log p(\mathbf{x}_t)]dt + g(t)d\bar{\mathbf{w}} \tag{2}$$

Where the $\bar{w}$ is the standard Brownian motion in reverse time. In practice, a neural network (Usually a U-Net) $s(x_t, t; \boldsymbol{\theta})$ is used to estimate the score $\nabla_{\mathbf{x}_t} \log p(\mathbf{x}_t)$ for each time step, thereby approximating the reverse SDE. The optimization objective of the model can be written as:

$$\boldsymbol{\theta}^* = \arg \min_{\boldsymbol{\theta}} \mathbb{E}_{t \sim U(0,T)} \mathbb{E}_{\mathbf{x}_0 \sim p(\mathbf{x}_0)} \mathbb{E}_{\mathbf{x}_t \sim p(\mathbf{x}_t|\mathbf{x}_0)} [\lambda(t) \| s(\mathbf{x}_t, t; \boldsymbol{\theta}) - \nabla_{\mathbf{x}_t} \log p(\mathbf{x}_t|\mathbf{x}_0) \|^2] \tag{3}$$

Where $\lambda : [0, T] \to \mathbb{R}_+$ is a weighting function with respect to time, $\nabla_{\mathbf{x}_t} \log p(\mathbf{x}_t|\mathbf{x}_0)$ can be obtained through the transition kernel of the forward process. Given sufficient data and model capacity, the converged model $s(\mathbf{x}_t, t; \boldsymbol{\theta}^*)$ matches $\nabla_{\mathbf{x}_t} \log p(\mathbf{x}_t)$ for almost all $\mathbf{x}_t$ (Song et al., 2020).

## 3.2 PROBLEM FORMULATION

In this paper, we aim to to improve the detector's recognition of new agricultural scenes in agriculture with limited labeled data. Assume $\mathcal{D}^{(1)} = \{\mathbf{x}^{(1)}, \mathbf{y}_1^{(1)}, \mathbf{y}_2^{(1)}\}$ is an existing object detection dataset, where $\mathbf{x}^{(1)}$ represents all the images in $\mathcal{D}^{(1)}$, $\mathbf{y}_1^{(1)}$ and $\mathbf{y}_2^{(1)}$ represents the domain information and the bounding box annotations of $\mathbf{x}^{(1)}$, respectively. $\mathbf{x}^{(2)}, \mathbf{y}_1^{(2)}$ are images from the new scenes and their corresponding domain information. Because $\mathbf{y}_1^{(1)} \neq \mathbf{y}_1^{(2)}$, the detectors trained on $\mathcal{D}^{(1)}$ may not able to recognize $\mathbf{x}^{(2)}$. We expect to leverage diffusion to build a synthetic dataset $\hat{\mathcal{D}}^{(2)} = \{\hat{\mathbf{x}}^{(2)}, \mathbf{y}_1^{(2)}, \hat{\mathbf{y}}_2^{(2)}\}$, and improve detectors' recognition of $\mathbf{x}^{(2)}$ by fine-tuning on $\hat{\mathcal{D}}^{(2)}$.

First, the images generated by the diffusion model should align with the context of the target domain. This requires the diffusion model to distinguish between different domains and sample from $p(\mathbf{x}|\mathbf{y}_1^{(2)})$. Given that $\hat{\mathbf{x}}^{(2)}$ is expected to resemble $\mathbf{x}^{(2)}$, the common approach for constructing synthetic datasets, synthesizing $\hat{\mathbf{x}} \sim p(\mathbf{x})$ first, then obtaining labels with an off-the-shelf model (Li et al., 2023b; Zhang et al., 2023b; Kim et al., 2024), cannot be applied. To generate detection data for the new domains, the diffusion model should be able to sample from $p(\mathbf{x}|\mathbf{y}_1^{(2)}, \hat{\mathbf{y}}_2^{(2)})$.

## 3.3 INCORPORATING DOMAIN EMBEDDING FOR DOMAIN-AWARE IMAGE GENERATION

We expect to incorporate external domain embeddings to decouple the learning of domain-specific features from diffusion, thus enabling domain-aware image generation. These domain embeddings should satisfy two key criteria: 1. They must effectively guide the diffusion model to generate images that align with the target domain. This alignment should be at both the image and instance level. 2. They should be easily obtainable for various domains, including unseen ones.

Ma et al. (2022) demonstrated that combined instance and image level DA produces better results than either method alone. This suggests that the features extracted by the model encompass both image-level and instance-level domain characteristics, which is exactly what we want! Furthermore, David et al. (2021) used a ResNet (He et al., 2016) trained on ImageNet to extract features from the GWHD dataset, to suggest that dimensionality reduction can distinguish training and test set features. Similarly, as shown in Fig. 2a, our finer-grained test indicates that simple dimensionality reduction can differentiate features from different domains.This suggests that domain-specific features reflect unique domain characteristics, even without prior training on those domains.

Based on the above, we propose using a pre-trained vision encoder to extract features as domain embeddings. The embeddings can then be integrated into the U-Net via cross-attention:

$$\text{Attention}(Q, K, V) = \text{softmax}\left(\frac{QK^\top}{\sqrt{d}}\right) V = \text{softmax}\left(\frac{Q(W^K f_d(x_{ref}))^\top}{\sqrt{d}}\right) W^V f_d(x_{ref}) \tag{4}$$

Where the $W^K \in \mathbb{R}^{d_1 \times d_2}$ and $W^V \in \mathbb{R}^{d_1 \times d_2}$ are projection matrices that convert domain embeddings into Key and Value features, and $f_d : \mathbb{R}^{h_1 \times w_1 \times c_1} \to \mathbb{R}^{d_1}$ is the domain encoder used to obtain domain embedding for each domain reference image $x_{ref}$. By default, we use a ViT-B (Dosovitskiy et al., 2020) pre-trained with CLIP (Radford et al., 2021) as the domain encoder.

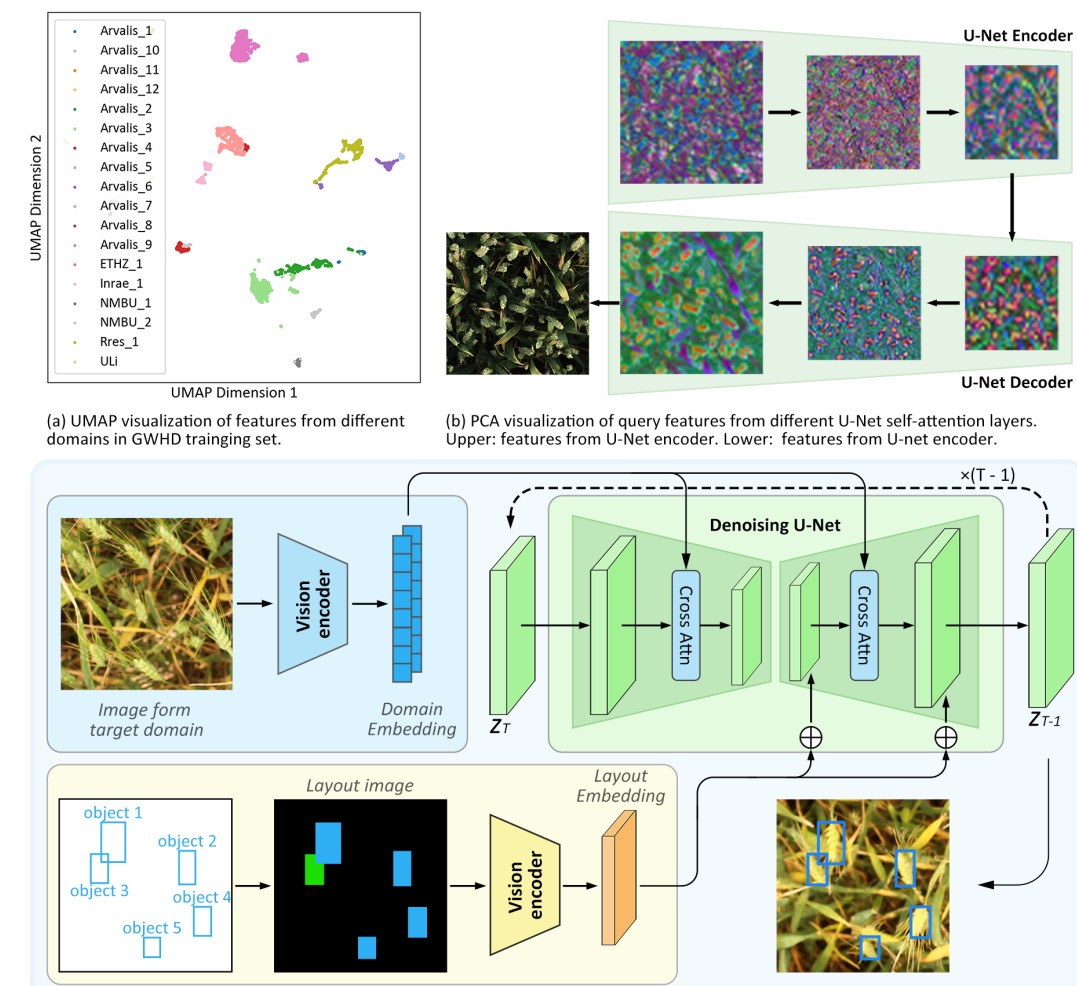

(a) UMAP visualization of features from different domains in GWHD trainging set.

(b) PCA visualization of query features from different U-Net self-attention layers. Upper: features from U-Net encoder. Lower: features from U-net encoder.

(c) The overall structure of DODA.

Figure 2: (a) Visualization of the image features from the GWHD training set. The image features are extracted by MAE (He et al., 2022a) and different subdomains are distinguishable by color. (b) Features in shallow layers are relatively noisy, while deeper layers progressively form a clearer layout of the image. (c) The architecture of DODA. Upper left: a pre-trained vision encoder provides domain features, enabling image generation for target domains without additional training. Lower left: our layout-image-to-image (LI2I) method directly represents layouts as images and encodes them using a vision encoder. This preserves the spatial relationships and structure of the layout.

**Decoupling layout information from domain embedding.** The features extracted by the vision encoder contain both domain-specific features and layout information. The layout information can interfere with controlling the layout of the generated images. To decouple layout information from domain embedding, we employ a simple, yet effective method termed asymmetric augmentation. In asymmetric augmentation, both the domain reference image and the denoising target image are obtained by transforming the original image during training, but the augmentation of the reference image is relatively weaker. More details can be found in Appendix B.

### 3.4 ENCODING LAYOUT IMAGES WITH VISION MODEL FOR SIMPLER AND BETTER ALIGNMENT

Generating images from layouts is a key focus in image generation. Existing methods (Zheng et al., 2023; Chen et al., 2023; Rombach et al., 2022; Li et al., 2023a) first represent the layout as text, and encode it with a language model, then fuse the layout embedding into the diffusion model through

cross-attention. We refer to these methods collectively as LT2I (layout-text-to-image). From our perspective, LT2I disrupts the layout's spatial relationships in multiple ways: 1. Since, LT2I represents the layout as text, which is embedded as discrete tokens before being fed into the transformer, The transformer is forced to learn to reconstruct the spatial relationships from the discretized tokens. 2. Before cross-attention, features from the diffusion model are flattened, which degrades spatial information and makes alignment more difficult.

We propose to address these problems by simplifying the process. As shown in the lower left of Fig. 2c, inspired by (Zhang et al., 2023a; Mou et al., 2024), we represent the layout as image, which naturally has accurate spatial information, eliminating the need of learning. The layout encoder then interprets the hierarchical and positional relationships between bounding boxes, and converts rectangles into object shapes. Finally, we fuse the features through addition to maximally retain spatial information. Corresponding to LT2I, we refer to this method as LI2I (layout-image-to-image).

**Channel Coding for Overlapped Instances.** Bounding boxes will inevitably overlap with each other, so to help the layout encoder distinguish instances, we assign different colors to overlapped instances. Specifically, we represent the overlap relationships of the bounding boxes in each image as an adjacency matrix, and use Alg. 1 to arrange the boxes.

---

**Algorithm 1** Bounding Boxes Arrangement

---

**Input**: Adjacency matrix $A$ of bounding boxes, the number of bounding boxes $n$.
**Output**: Array $channels$ containing the assigned channel for each bounding box.

1: $channels \leftarrow$ array of length $n$ initialized with 0
2: **for** $i = 1$ to $n$ **do**
3:     $C_i \leftarrow \emptyset$
4:     **for** $j = 1$ to $i$ **do**
5:        **if** $A[i][j] = 1$ and $channels[j] \neq 0$ **then**
6:           Add $channels[j]$ to $C_i$
7:        **end if**
8:     **end for**
9:     Assign the smallest channel not in $C_i$ to $channels[i]$
10: **end for**
**Return** $channels$

---

**Design of Layout Encoder.** Our layout encoder has a simple structure, as it does not need to convert the discrete input to spatial relationships. The layout encoder consists of a stack of time-dependent residual layers and downsampling layers. The output of each residual layer is $f_{res}(\mathbf{a}, t) + \mathbf{a}$, here $\mathbf{a}$ is the output of last layer, and $t$ is the timestep.

**Layers to Merge Layout Embeddings.** We observed that shallow U-Net layers produce noisy, localized features. As the layers depend, the features become increasingly abstract and holistic, gradually forming the overall layout of the image (shown in Fig. 2b). Therefore, we propose to merge the layout embeddings with the features of deeper layers (layers in the U-Net decoder) to better convey the layout information.

### 3.5 Unified Optimization Objective for Multi-Conditional Diffusion

Based on the assumption that the diffusion model can learn to utilize conditions during training, thereby generating images $\hat{\mathbf{x}} \sim p(\mathbf{x}|\mathbf{y}_1, \mathbf{y}_2, \ldots, \mathbf{y}_n)$, many studies integrate multiple conditions, e.g., text, depth, pose, into the training of diffusion model (Li et al., 2023b; Chen et al., 2023; Lu et al., 2023):

$$\boldsymbol{\theta}^* = \arg\min_{\boldsymbol{\theta}} \mathbb{E}_{t \sim U(0,T)} \mathbb{E}_{\mathbf{x}_0, \mathbf{y}_1, \mathbf{y}_2 \sim p(\mathbf{x}_0, \mathbf{y}_1, \mathbf{y}_2)} \mathbb{E}_{\mathbf{x}_t \sim p(\mathbf{x}_t|\mathbf{x}_0)} [\lambda(t) \| s(\mathbf{x}_t, \mathbf{y}_1, \mathbf{y}_2, t; \boldsymbol{\theta}) - \nabla_{\mathbf{x}_t} \log p(\mathbf{x}_t|\mathbf{x}_0) \|^2] \quad (5)$$

However, these studies simply incorporate multiple conditions into the diffusion model without modifying the optimization objective, making it unclear whether $\hat{\mathbf{x}} \sim p(\mathbf{x}|\mathbf{y}_1, \mathbf{y}_2, \ldots, \mathbf{y}_n)$ is actually achieved. To address this, we derive the two-conditional optimization objective as follows.

First, applying the forward diffusion process in in Eq. 1, obtains the perturbed distribution $p(\mathbf{x}_t|\mathbf{y}_1, \mathbf{y}_2)$, according to Anderson (1982), the corresponding reverse-time SDE is given by:

$$d\mathbf{x} = [f(\mathbf{x}, t) - g(t)^2 \nabla_{\mathbf{x}_t} \log p(\mathbf{x}_t|\mathbf{y}_1, \mathbf{y}_2)]dt + g(t)d\bar{\mathbf{w}} \quad (6)$$

By simulating Eq. 6, we can generate samples from $p(\mathbf{x}_t|\mathbf{y}_1, \mathbf{y}_2)$. To construct the this reverse time SDE, we need to estimate the conditional score, similar to Eq. 3, the training objective is:

$$\boldsymbol{\theta}^* = \arg\min_{\boldsymbol{\theta}} \mathbb{E}_{t \sim U(0,T)} \mathbb{E}_{\mathbf{x}_t, \mathbf{y}_1, \mathbf{y}_2 \sim p(\mathbf{x}_t, \mathbf{y}_1, \mathbf{y}_2)} [\lambda(t) \| s(\mathbf{x}_t, \mathbf{y}_1, \mathbf{y}_2, t; \boldsymbol{\theta}) - \nabla_{\mathbf{x}_t} \log p(\mathbf{x}_t|\mathbf{y}_1, \mathbf{y}_2) \|^2] \quad (7)$$

However, the $\nabla_{\mathbf{x}_t} \log p(\mathbf{x}_t|\mathbf{y}_1, \mathbf{y}_2)$ in Eq. 7 is hard to access. Batzolis et al. (2021) provided a method to approximate $\nabla_{\mathbf{x}_t} \log p(\mathbf{x}_t|\mathbf{y})$. By generalizing it to multi-conditional setting, we prove that the optimal solution of Eq. 7 is the same as the solution of Eq. 5 (Proof in Appendix. A):

**Proposition 1.** *The solution that minimizes*
$\mathbb{E}_{t \sim U(0,T)} \mathbb{E}_{\mathbf{x}_0, \mathbf{y}_1, \mathbf{y}_2 \sim p(\mathbf{x}_0, \mathbf{y}_1, \mathbf{y}_2)} \mathbb{E}_{\mathbf{x}_t \sim p(\mathbf{x}_t|\mathbf{x}_0)} [\lambda(t) \| s(\mathbf{x}_t, \mathbf{y}_1, \mathbf{y}_2, t; \boldsymbol{\theta}) - \nabla_{\mathbf{x}_t} \log p(\mathbf{x}_t|\mathbf{x}_0) \|^2]$
*is the same as the solution minimizes*
$\mathbb{E}_{t \sim U(0,T)} \mathbb{E}_{\mathbf{x}_t, \mathbf{y}_1, \mathbf{y}_2 \sim p(\mathbf{x}_t, \mathbf{y}_1, \mathbf{y}_2)} [\lambda(t) \| s(\mathbf{x}_t, \mathbf{y}_1, \mathbf{y}_2, t; \boldsymbol{\theta}) - \nabla_{\mathbf{x}_t} \log p(\mathbf{x}_t|\mathbf{y}_1, \mathbf{y}_2) \|^2]$

With this Proposition, we have established that the optimal solution $s(\mathbf{x}_t, \mathbf{y}_1, \mathbf{y}_2, t; \boldsymbol{\theta}^*)$ of Eq. 5 is able to approximate the multi-conditional score $\nabla_{\mathbf{x}_t} \log p(\mathbf{x}_t|\mathbf{y}_1, \mathbf{y}_2)$.

**Two-stage Training.** In practice, the box annotations $\mathbf{y}_2^{(1)}$ in $\mathcal{D}^{(1)}$ are very limited, while $\mathbf{x}$ and $\mathbf{y}_1$ are relatively easy to obtain. Therefore, we suggest building a larger dataset without box annotations $\mathcal{D}^{(3)} = \{\mathbf{x}^{(3)}, \mathbf{y}_1^{(3)}\}$, where $\mathbf{x}^{(1)} \subset \mathbf{x}^{(3)}$, $\mathbf{y}_1^{(3)} = f_d(\mathbf{x}^{(3)})$, and perform pre-training on $\mathcal{D}^{(1)}$ to achieve a better estimation of $p(\mathbf{x}|\mathbf{y}_1)$, the training objective can be written as:

$$\boldsymbol{\theta}^* = \arg\min_{\boldsymbol{\theta}} \mathbb{E}_{t \sim U(0,T)} \mathbb{E}_{\mathbf{x}_0, \mathbf{y}_1 \sim p(\mathbf{x}_0, \mathbf{y}_1)} \mathbb{E}_{\mathbf{x}_t \sim p(\mathbf{x}_t|\mathbf{x}_0)} [\lambda(t) \| s(\mathbf{x}_t, \mathbf{y}_1, t; \boldsymbol{\theta}) - \nabla_{\mathbf{x}_t} \log p(\mathbf{x}_t|\mathbf{x}_0) \|^2] \quad (8)$$

After pre-training, post-training can be conducted using object detection dataset $\mathcal{D}^{(3)}$ with the objective of Eq. 5. The effectiveness of this two-stage training process is shown in Sec. 4.2.

# 4 EXPERIMENT

In Sec. 4.1.1, we test how effectively DODA could adapt the detector to new agricultural domains. In Sec. 4.1.2, we compare our proposed LI2I method with previous L2I methods. Lastly, in Sec. 4.2, we conduct ablation studies to understand the impact of the proposed components and experiment settings. By default, DODA employs latent diffusion (LDM) (Rombach et al., 2022) as the base diffusion model. The pre-training of DODA is performed on the GWHD 2+ dataset, followed by post-training on the GWHD training set. The maximum number of channels for the layout image is 3. Hyperparameters for training and more implementation details can be found in Appendix B.

We use Fréchet Inception Distance (FID), Inception Score (IS), COCO Metrics, YOLO Score and Feature Similarity (FS) as evaluation metrics. Their specific definitions can be found in Appendix C.

## 4.1 MAIN RESULTS

### 4.1.1 SYNTHETIC DATA FOR AGRICULTURAL OBJECT DETECTION DOMAIN ADAPTATION

**Setup.** We initialize a YOLOX-L (Ge et al., 2021) with COCO pre-trained weights, train it on the GWHD training set, and use this as the baseline and base model. To evaluate the effectiveness of DODA for domain adaptation, we focus on the domains within the GWHD test set where $AP_{50}$ lower than 0.8. For each domain, we use DODA to generate a 200 image dataset, then fine-tune the YOLOX-L on this synthetic data for one epoch.

Samples of data generated by DODA for different agricultural domains can be seen in the Fig. 1 left. As shown in Table 1, the data of 13 domains were collected from different devices, different regions and different stages of wheat head development. After fine-tuning the detector with domain-specific data synthesized by DODA, recognition across these domains improved, the AP increased to 15.6, with an average improvement of 7.5.

In addition to the GWHD, we test our method on wheat images collected by UAV, which have significantly different spatial resolutions. As shown in the Table 1, our method can effectively help the detector adapt to UAV image data. Furthermore, we explore cross-crop adaption. As highlighted in the last row of Table 1, our method successfully adapts the detector trained on wheat to sorghum (Ghosal et al., 2019). These results suggest our method effectively helps the detector adapt to new

Table 1: Domain-specific performance on GWHD test set after fine-tuning with DODA-generated data. Improvements over baseline are marked in red. Consistent improvements across various domains demonstrate that DODA is effective in adapting detectors to new agricultural domains.

| Domain | AP | $AP_{50}$ | $AP_{75}$ | $AP^s$ | $AP^m$ | $AP^l$ | Development stage | Platform | Country |
|---|---|---|---|---|---|---|---|---|---|
| Global Wheat Head Detection | | | | | | | | | |
| ARC_1 | 35.0 | 29.7 | 8.1 | 35.0 | 39.9 | 73.1 | Filling | Handheld | Sudan |
| + Ours | 37.4 (+2.4) | 78.3 (+5.2) | 32.3 (+2.6) | 8.3 (+0.2) | 36.8 (+1.8) | 43.0 (+3.1) | | | |
| CIMMYT_1 | 26.7 | 65.7 | 16.9 | 5.3 | 24.6 | 45.9 | Postflowering | Cart | Mexico |
| + Ours | 40.4 (+13.7) | 80.7 (+15.0) | 36.1 (+19.2) | 16.2 (+10.9) | 39.2 (+14.6) | 55.5 (+9.6) | | | |
| KSU_1 | 37.1 | 72.0 | 34.9 | 9.5 | 40.1 | 51.5 | Postflowering | Tractor | US |
| + Ours | 46.7 (+9.6) | 84.7 (+12.7) | 47.2 (+12.3) | 22.0 (+12.5) | 49.1 (+9.0) | 53.4 (+1.9) | | | |
| KSU_2 | 34.9 | 74.0 | 29.6 | 6.9 | 38.3 | 58.7 | Postflowering | Tractor | US |
| + Ours | 42.2 (+7.3) | 86.5 (+12.5) | 33.9 (+4.3) | 16.2 (+9.3) | 44.9 (+6.6) | 61.5 (+2.8) | | | |
| KSU_3 | 27.4 | 67.8 | 16.2 | 6.0 | 26.5 | 42.6 | Filling | Tractor | US |
| + Ours | 39.2 (+11.8) | 81.3 (+13.5) | 31.8 (+15.6) | 16.9 (+10.9) | 39.0 (+12.5) | 50.0 (+7.4) | | | |
| KSU_4 | 22.7 | 56.3 | 14.6 | 1.2 | 22.6 | 40.6 | Ripening | Tractor | US |
| + Ours | 38.3 (+15.6) | 75.1 (+18.8) | 34.1 (+19.5) | 10.0 (+8.8) | 39.3 (+16.7) | 49.4 (+8.8) | | | |
| Terraref_1 | 11.3 | 33.1 | 5.1 | 1.0 | 13.1 | 43.6 | Ripening | Gantry | US |
| + Ours | 20.7 (+9.4) | 54.6 (+21.5) | 10.6 (+5.5) | 4.1 (+3.1) | 24.0 (+10.9) | 43.2 (-0.4) | | | |
| Terraref_2 | 9.1 | 23.7 | 5.2 | 0.6 | 12.3 | 31.7 | Filling | Gantry | US |
| + Ours | 16.4 (+7.3) | 41.5 (+17.8) | 10.2 (+5.0) | 2.7 (+2.1) | 20.7 (+8.4) | 47.2 (+15.5) | | | |
| Ukyoto_1 | 35.0 | 68.4 | 31.8 | 4.9 | 38.7 | 56.8 | Postflowering | Handheld | Japan |
| + Ours | 35.6 (+0.6) | 70.1 (+1.7) | 32.6 (+0.8) | 5.5 (+0.6) | 39.0 (+0.3) | 57.3 (+0.5) | | | |
| UQ_8 | 31.3 | 66.3 | 24.8 | 12.8 | 39.1 | 53.3 | Ripening | Handheld | Australia |
| + Ours | 36.3 (+5.0) | 70.3 (+4.0) | 33.6 (+8.8) | 16.6 (+3.8) | 44.4 (+5.3) | 58.3 (+5.0) | | | |
| UQ_9 | 30.9 | 66.8 | 25.6 | 8.4 | 35.0 | 54.4 | Filling-ripening | Handheld | Australia |
| + Ours | 36.9 (+6.0) | 72.3 (+5.5) | 34.8 (+9.2) | 14.6 (+6.2) | 41.0 (+6.0) | 58.4 (+4.0) | | | |
| UQ_10 | 37.7 | 78.5 | 31.2 | 20.7 | 43.7 | 53.8 | Filling-ripening | Handheld | Australia |
| + Ours | 43.3 (+5.6) | 81.5 (+3.0) | 41.1 (+9.9) | 26.5 (+5.8) | 49.0 (+5.3) | 56.6 (+2.8) | | | |
| UQ_11 | 27.8 | 69.5 | 16.4 | 17.0 | 34.0 | 42.0 | Postflowering | Handheld | Australia |
| + Ours | 31.0 (+3.2) | 71.6 (+2.1) | 21.6 (+5.2) | 21.2 (+4.2) | 36.2 (+2.2) | 45.7 (+3.7) | | | |
| K | 13.3 | 35.5 | 6.6 | 9.0 | 18.0 | - | Ripening | UAV | - |
| + Ours | 28.2 (+14.9) | 54.9 (+19.4) | 25.9 (+19.3) | 20.0 (+11.0) | 37.5 (+19.5) | - | | | |
| Sorghum | 17.3 | 40.0 | 12.6 | 17.7 | 21.5 | - | Ripening | UAV | Australia |
| + Ours | 29.4 (+12.1) | 70.5 (+30.5) | 17.9 (+5.3) | 30.0 (+12.3) | 31.5 (+10.0) | - | | | |

scenes of agricultural field, bridging the gap between limited manual annotations and complex, ever-changing agricultural environments.

Notably, DODA employs a single model to assist the detector in adapting to multiple domains. The cross domain capability of the DODA highlights its potential to lower the technical and financial barriers to using object detection.

### 4.1.2 COMPARISONS WITH PREVIOUS LAYOUT-TO-IMAGE METHODS

**Setup.** Consistent with previous L2I studies, we perform experiments on the COCO dataset to demonstrate the effectiveness of our proposed method. Following the setting of Chen et al. (2023); Cheng et al. (2023), we apply the proposed LI2I method to Stable Diffusion (Rombach et al., 2022) v1.5. To preserve the knowledge learned from billions of images (Schuhmann et al., 2022), we use the encoder of U-Net as the layout encoder, and following Zhang et al. (2023a) initialize it with the weight of the diffusion model. Since Stable Diffusion is a T2I model, we constructed a simple text prompt for our method: "a photograph with $(N_{cls}^1)(Cls^1), \ldots, (N_{cls}^i)(Cls^i)$", where $Cls^i$ is the category, and $N_{cls}^i$ denotes the number of objects belonging to that category. Since COCO contains multiple categories, we design a layout coding method that different from Sec. 3.4, objects of the same category are depicted with the same hue but weaker brightness, and the bounding box of each object is drawn in descending order of area.

Table 2 shows that our LI2I method significantly outperforms all previous L2I methods in terms of controllability (mAP), while maintaining high image quality (FID) and diversity (IS). Moreover, in contrast to previous methods that represent layouts as text (LayoutDiffusion, Layout diffuse, GeoDiffusion), our layout images technique overcomes limitations of text-based layouts, allowing for more precise and detailed control, including small objects that were previously challenging to produce ($AP^s$). ControlNet Zhang et al. (2023a) takes semantic masks as conditions, and can achieve

Table 2: Quantitative results on COCO-val2017. Our proposed layout-image-to-image (LI2I) method achieves significant improvements in controllability compared to previous works, closely approximating the results of real images, while also maintaining high image quality and diversity. (Filter data: filtering objects whose area ratio is less than 0.02, and images with more than 8 objects.)

| Method | Filter data | YOLO Score↑ | | | | | | FID↓ | IS↑ |
|---|---|---|---|---|---|---|---|---|---|
| | | mAP | $AP_{50}$ | $AP_{75}$ | $AP^s$ | $AP^m$ | $AP^l$ | | |
| $256 \times 256$ | | | | | | | | | |
| Real image | ✓ | 55.5 | 70.7 | 60.8 | - | 51.2 | 69.0 | - | - |
| PLGAN (Wang et al., 2022a) | ✓ | 21.4 | 35.2 | 22.9 | - | 16.8 | 27.3 | 35.9 | 17.7±0.9 |
| LostGANv2 (Sun & Wu, 2021) | ✓ | 26.6 | 41.6 | 28.3 | - | 21.9 | 34.3 | 37.0 | 17.0±0.9 |
| LAMA (Li et al., 2021b) | ✓ | 38.3 | 53.9 | 42.3 | - | 34.8 | 45.0 | 37.5 | 18.4±1.0 |
| LayoutDiffusion (Zheng et al., 2023) | ✓ | 30.6 | 56.6 | 29.5 | - | 20.0 | 43.4 | **23.6** | 24.3±1.2 |
| **LI2I (ours)** | ✓ | **54.3** | **72.1** | **59.1** | - | **48.7** | **59.9** | 31.5 | **24.6±1.2** |
| Real image | - | 35.5 | 51.2 | 37.5 | 15.3 | 48.3 | 62.2 | - | 29.0±1.3 |
| LayoutDiffusion (Zheng et al., 2023) | - | 6.0 | 14.9 | 3.8 | 0.2 | 4.7 | 19.8 | **20.5** | 21.8±1.1 |
| GeoDiffusion (Chen et al., 2023) | - | 27.3 | 38.5 | 29.3 | 2.8 | 40.3 | **63.2** | 34.3 | 24.7±1.1 |
| ControlNet M2I (Zhang et al., 2023a) | - | 29.4 | 39.4 | 31.1 | 11.2 | 39.3 | 52.4 | 49.1 | 18.4±0.9 |
| **LI2I (ours)** | - | **31.8** | **45.4** | **33.0** | **12.3** | **41.5** | 57.2 | 29.9 | **28.5±0.8** |
| $512 \times 512$ | | | | | | | | | |
| Real image | - | 45.2 | 63.3 | 48.5 | 17.9 | 45.1 | 61.5 | - | 31.5±1.2 |
| Layout diffuse (Cheng et al., 2023) | ✓ | 4.2 | 11.3 | 2.3 | - | 0.1 | 4.6 | 33.5 | **29.6±1.1** |
| GeoDiffusion (Chen et al., 2023) | - | 27.7 | 40.7 | 29.6 | 0 | 13.0 | 57.8 | 28.8 | 26.4±2.4 |
| ControlNet M2I (Zhang et al., 2023a) | - | 39.5 | 50.3 | 41.6 | 16.6 | 39.0 | 52.2 | 46.9 | 20.2±0.9 |
| **LI2I (ours)** | - | **42.5** | **56.1** | **44.9** | **16.1** | **40.9** | **59.1** | **24.9** | 29.4±1.1 |

relatively accurate layout control, but it imposes stricter constraints, leading to much lower image quality and diversity. More importantly, using algorithms to obtain masks is challenging, making M2I unsuitable for data generation. The quantitative comparison of above methods can be found in Appendix G.

We report the YOLO Score on the real validation set in Table 2 as the strong baseline. The YOLO Score achieved on our generated images closely approximate those obtained from real images. This close alignment highlights the accuracy of our synthetic labels, and indicates significant progress in narrowing the gap between synthetic and real data.

Among methods (Layout diffuse, GeoDiffusion) based on latent diffusion (Rombach et al., 2022), our method achieves the lowest FID. However, the FID is still relatively high compared to LayoutDiffusion, which does not use the variational autoencoder (Kingma & Welling, 2013). To enhance the quality of the generated data, image quality needs further improvement.

## 4.2 ABLATION STUDY

In this section, we perform ablation studies to evaluate the impact of each design and setting of the proposed method, additional results are in Appendix E. For synthetic data quality (AP), we focus on the two most challenging domains in the GWHD test set: "Terraref1" and "Terraref2". Since they are relatively similar, we combine these into a single domain, "Terraref". To evaluate L2I controllability (YOLO Score) and image quality (FID and FS), we randomly select 5,000 images from the test set.

**Scaling dataset and GWHD 2+.** The separation of domain features by the domain encoder enables us to pre-train the diffusion model using a larger set of images, without requiring labels. In Table 3, we explore the effect of the proposed two-stage training. Specifically, in order to evaluate the impact of pre-training dataset size we randomly sampled 0%, 50% (12k images) and 100% (23k images) of unlabeled data from the GWHD. We consider two scenarios: target domain images are accessible (**w/ target images** in table) or inaccessible (**w/o target images**) during pre-traing. After pre-training, we generate a dataset of 400 images for "Terraref".

As shown in Table 3, when no additional unlabeled data is used, training is one-stage, leading to the worst results. As the size of the pre-training dataset increases, the $AP_{50}$ steadily improves, demonstrating the effectiveness of the two-stage training process. Notably, pre-training diffusion

Table 3: Impact of pre-training dataset size on the quality of generated data, measured by $AP_{50}$. Pre-training diffusion models with more unlabeled images, especially those from the target domain, can improve data quality.

| Dataset Size | w/o target images | w/ target images |
|---|---|---|
| 33k | 41.1 | 41.1 |
| 45k | 44.8 | 45.4 |
| 56k | 45.4 | 49.6 |
| 121k | 48.4 | 50.7 |

Table 4: Ablations on domain encoder. Various pre-trained vision models can serve as domain encoders, CLIP performs best.

| Domain Encoder | FS↑ |
|---|---|
| ✗ | 0.477 |
| CLIP | **0.769** |
| MAE | 0.747 |
| ResNet101 | 0.751 |

Table 5: Features from different layers in MAE as domain embedding.

| Layers | FS↑ | FID↓ |
|---|---|---|
| 2 | 0.622 | 44.8 |
| 4 | 0.664 | 37.7 |
| 8 | 0.719 | 30.6 |
| 12 | 0.747 | 28.0 |

Table 6: Ablations on the optimal position to merge the layout embedding.

| Encoder | Decoder | YOLO Score↑ | | | | | | FID↓ |
|---|---|---|---|---|---|---|---|---|
| | | mAP | $AP_{50}$ | $AP_{75}$ | $AP^s$ | $AP^m$ | $AP^l$ | |
| ✗ | ✓ | 23.1 | 64.1 | 10.2 | 17.1 | 28.2 | 22.0 | 27.2 |
| ✓ | ✗ | 26.0 | 69.5 | 12.5 | 20.5 | 30.7 | 21.8 | 27.7 |
| ✓ | ✓ | 25.3 | 67.5 | 11.9 | 18.8 | 30.7 | 23.1 | 27.3 |

models with unlabeled images from the target domain significantly enhances data quality. To support pre-training, we introduce GWHD 2+, an extension of GWHD with 65k additional unlabeled wheat images from 12 domains. As shown in last row of Table 3, GWHD 2+ further improves the data quality and narrows the gap between the scenarios with and without access to target domain images. The performance gains from pre-training with additional unlabeled images, particularly those from the target domain, suggest that our GWHD 2+ dataset is still insufficiently large. A priority for the future is to collect more images to enhance DODA's ability.

**Domain encoder.** Table 4 shows FS with and without the domain encoder. We also train DODA using MAE (He et al., 2022a) and ResNet101 (He et al., 2016) as the domain encoder. Without domain encoder, diffusion randomly samples from the training set, leading to lower FS. Independently of the architecture and training data, various pre-trained vision models can guide diffusion to generate image with specific features. Using ViT as the backbone, CLIP performs the best, while MAE worse than ResNet. Compared to contrastive learning, MAE focuses more on high-frequency texture features (Park et al., 2023; Vanyan et al., 2023), which we hypothesize affects the quality of the domain embedding. To explore this, we test features from different MAE layers, as shallow layers are generally associated with high-frequency, low-level features. As shown in Table 5, using shallower MAE features further impacts image quality.

**Position to merge the layout embedding.** Table 6 presents the impact of layout embedding fusion position on layout controllability. It can be seen that fusing layout embedding with the layers of denoising U-Net decoder can more effectively convey layout information.

## 5 CONCLUSION

This paper presents DODA, a framework that incorporates domain features and image layout conditions to extend a diffusion model, enabling it to generate detection data for new agricultural domains. With just a few reference images from the target domain, DODA can generate data for it without additional training. Extensive experiments demonstrated the effectiveness of DODA-generated data in adapting detectors to diverse agricultural domains, as demonstrated by significant AP improvements across multiple domains. The simplicity and effectiveness reduce barriers for more growers to use object detection for their personalized scenarios.

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

APPENDIX

## A    PROOF

**Proposition 1.** *The solution that minimizes*

$\mathbb{E}_{t \sim U(0,T)} \mathbb{E}_{\mathbf{x}_0,\mathbf{y}_1,\mathbf{y}_2 \sim p(\mathbf{x}_0,\mathbf{y}_1,\mathbf{y}_2)} \mathbb{E}_{\mathbf{x}_t \sim p(\mathbf{x}_t|\mathbf{x}_0)} [\lambda(t) \| s(\mathbf{x}_t,\mathbf{y}_1,\mathbf{y}_2,t;\boldsymbol{\theta}) - \nabla_{\mathbf{x}_t} \log p(\mathbf{x}_t|\mathbf{x}_0) \|^2]$

*is the same as the solution minimizes*

$\mathbb{E}_{t \sim U(0,T)} \mathbb{E}_{\mathbf{x}_t,\mathbf{y}_1,\mathbf{y}_2 \sim p(\mathbf{x}_t,\mathbf{y}_1,\mathbf{y}_2)} [\lambda(t) \| s(\mathbf{x}_t,\mathbf{y}_1,\mathbf{y}_2,t;\boldsymbol{\theta}) - \nabla_{\mathbf{x}_t} \log p(\mathbf{x}_t|\mathbf{y}_1,\mathbf{y}_2) \|^2]$

*Proof.* Let $f(\mathbf{x}_t,\mathbf{x}_0,\mathbf{y}_1,\mathbf{y}_2) := \lambda(t) \| s(\mathbf{x}_t,\mathbf{y}_1,\mathbf{y}_2,t;\boldsymbol{\theta}) - \nabla_{\mathbf{x}_t} \log p(\mathbf{x}_t|\mathbf{x}_0) \|^2$, first, according to the Law of Iterated Expectations, we have:

$$\mathbb{E}_{t \sim U(0,T)} \mathbb{E}_{\mathbf{x}_0,\mathbf{y}_1,\mathbf{y}_2 \sim p(\mathbf{x}_0,\mathbf{y}_1,\mathbf{y}_2)} \mathbb{E}_{\mathbf{x}_t \sim p(\mathbf{x}_t|\mathbf{x}_0)} [\lambda(t) \| s(\mathbf{x}_t,\mathbf{y}_1,\mathbf{y}_2,t;\boldsymbol{\theta}) - \nabla_{\mathbf{x}_t} \log p(\mathbf{x}_t|\mathbf{x}_0) \|^2]$$

$$= \mathbb{E}_{t \sim U(0,T)} \mathbb{E}_{\mathbf{y}_1,\mathbf{y}_2 \sim p(\mathbf{y}_1,\mathbf{y}_2)} \mathbb{E}_{\mathbf{x}_0 \sim p(\mathbf{x}_0|\mathbf{y}_1,\mathbf{y}_2)} \mathbb{E}_{\mathbf{x}_t \sim p(\mathbf{x}_t|\mathbf{x}_0)} [f(\mathbf{x}_t,\mathbf{x}_0,\mathbf{y}_1,\mathbf{y}_2)]$$

$$= \mathbb{E}_{t \sim U(0,T)} \mathbb{E}_{\mathbf{y}_2 \sim p(\mathbf{y}_2)} \mathbb{E}_{\mathbf{y}_1 \sim p(\mathbf{y}_1|\mathbf{y}_2)} \mathbb{E}_{\mathbf{x}_0 \sim p(\mathbf{x}_0|\mathbf{y}_1,\mathbf{y}_2)} \mathbb{E}_{\mathbf{x}_t \sim p(\mathbf{x}_t|\mathbf{x}_0)} [f(\mathbf{x}_t,\mathbf{x}_0,\mathbf{y}_1,\mathbf{y}_2)] \quad (9)$$

The $\mathbf{y}_1$ and $\mathbf{y}_2$ are independent of each other. Given $\mathbf{x}_0$, $\mathbf{y}_1$ and $\mathbf{y}_2$ are independent of $\mathbf{x}_t$. Let $g(\mathbf{x}_t,\mathbf{x}_0,\mathbf{y}_1,\mathbf{y}_2) := \lambda(t) \| s(\mathbf{x}_t,\mathbf{y}_1,\mathbf{y}_2,t;\boldsymbol{\theta}) - \nabla_{\mathbf{x}_t} \log p(\mathbf{x}_t|\mathbf{x}_0,\mathbf{y}_1,\mathbf{y}_2) \|^2$, Eq. 9 can be written as:

$$\mathbb{E}_{t \sim U(0,T)} \mathbb{E}_{\mathbf{y}_2 \sim p(\mathbf{y}_2)} \mathbb{E}_{\mathbf{y}_1 \sim p(\mathbf{y}_1)} \mathbb{E}_{\mathbf{x}_0 \sim p(\mathbf{x}_0|\mathbf{y}_1,\mathbf{y}_2)} \mathbb{E}_{\mathbf{x}_t \sim p(\mathbf{x}_t|\mathbf{x}_0)} [f(\mathbf{x}_t,\mathbf{x}_0,\mathbf{y}_1,\mathbf{y}_2)]$$

$$= \mathbb{E}_{t \sim U(0,T)} \mathbb{E}_{\mathbf{y}_2 \sim p(\mathbf{y}_2)} \mathbb{E}_{\mathbf{y}_1 \sim p(\mathbf{y}_1)} \mathbb{E}_{\mathbf{x}_0 \sim p(\mathbf{x}_0|\mathbf{y}_1,\mathbf{y}_2)} \mathbb{E}_{\mathbf{x}_t \sim p(\mathbf{x}_t|\mathbf{x}_0,\mathbf{y}_1,\mathbf{y}_2)} [g(\mathbf{x}_t,\mathbf{x}_0,\mathbf{y}_1,\mathbf{y}_2)] \quad (10)$$

Let $t$, $\mathbf{y}_1$ and $\mathbf{y}_2$ be arbitrary fixed values, then we can define $h(\mathbf{x}_t) := s(\mathbf{x}_t,\mathbf{y}_1,\mathbf{y}_2,t;\boldsymbol{\theta})$, $q(\mathbf{x}_0) := p(\mathbf{x}_0|\mathbf{y}_1,\mathbf{y}_2)$ and $q(\mathbf{x}_t|\mathbf{x}_0) := p(\mathbf{x}_t|\mathbf{x}_0,\mathbf{y}_1,\mathbf{y}_2)$, applying the Law of Iterated Expectations, we have:

$$\mathbb{E}_{\mathbf{x}_0 \sim p(\mathbf{x}_0|\mathbf{y}_1,\mathbf{y}_2)} \mathbb{E}_{\mathbf{x}_t \sim p(\mathbf{x}_t|\mathbf{x}_0,\mathbf{y}_1,\mathbf{y}_2)} [\lambda(t) \| s(\mathbf{x}_t,\mathbf{y}_1,\mathbf{y}_2,t;\boldsymbol{\theta}) - \nabla_{\mathbf{x}_t} \log p(\mathbf{x}_t|\mathbf{x}_0,\mathbf{y}_1,\mathbf{y}_2) \|^2]$$

$$= \mathbb{E}_{\mathbf{x}_0 \sim q(\mathbf{x}_0)} \mathbb{E}_{\mathbf{x}_t \sim q(\mathbf{x}_t|\mathbf{x}_0)} [\lambda(t) \| h(\mathbf{x}_t) - \nabla_{\mathbf{x}_t} \log q(\mathbf{x}_t|\mathbf{x}_0) \|^2]$$

$$= \mathbb{E}_{\mathbf{x}_t \sim q(\mathbf{x}_t)} [\lambda(t) \| h(\mathbf{x}_t) - \nabla_{\mathbf{x}_t} \log q(\mathbf{x}_t) \|^2]$$

$$= \mathbb{E}_{\mathbf{x}_t \sim p(\mathbf{x}_t|\mathbf{y}_1,\mathbf{y}_2)} [\lambda(t) \| s(\mathbf{x}_t,\mathbf{y}_1,\mathbf{y}_2,t;\boldsymbol{\theta}) - \nabla_{\mathbf{x}_t} \log p(\mathbf{x}_t|\mathbf{y}_1,\mathbf{y}_2) \|^2] \quad (11)$$

Since $t$, $\mathbf{y}_1$, $\mathbf{y}_2$ are arbitrary, Eq. 11 is true for all $t$, $\mathbf{y}_1$, $\mathbf{y}_2$, via Eq. 11 and the Law of Iterated Expectations, we can easily rewrite Eq. 10 as:

$$\mathbb{E}_{t \sim U(0,T)} \mathbb{E}_{\mathbf{y}_2 \sim p(\mathbf{y}_2)} \mathbb{E}_{\mathbf{y}_1 \sim p(\mathbf{y}_1)} \mathbb{E}_{\mathbf{x}_t \sim p(\mathbf{x}_t|\mathbf{y}_1,\mathbf{y}_2)} [\lambda(t) \| s(\mathbf{x}_t,\mathbf{y}_1,\mathbf{y}_2,t;\boldsymbol{\theta}) - \nabla_{\mathbf{x}_t} \log p(\mathbf{x}_t|\mathbf{y}_1,\mathbf{y}_2) \|^2]$$

$$= \mathbb{E}_{t \sim U(0,T)} \mathbb{E}_{\mathbf{x}_t,\mathbf{y}_1,\mathbf{y}_2 \sim p(\mathbf{x}_t,\mathbf{y}_1,\mathbf{y}_2)} [\lambda(t) \| s(\mathbf{x}_t,\mathbf{y}_1,\mathbf{y}_2,t;\boldsymbol{\theta}) - \nabla_{\mathbf{x}_t} \log p(\mathbf{x}_t|\mathbf{y}_1,\mathbf{y}_2) \|^2] \quad (12)$$

$\square$

## B    IMPLEMENTATION DETAILS & HYPERPARAMETERS

In the GWHD, the images have high resolution but are relatively few in number. Therefore, we divided the original $1024 \times 1024$ images into 9 images of size $512 \times 512$ with step size 256. After splitting, there are a total of 58,635 images in GWHD. For the COCO 2017 dataset, we train with the official training set and test the proposed L2I method on the validation set.

By default, we use 4 NVIDIAA-V100-32GB, but all models in this paper can be trained on one single V100, and the GPU Memory usage and approximate computational requirements for one GPU are provided in the last two rows of Table 7 and Table 8. When training with multiple cards, all parameters including Learning Rate are the same except Iterations

## C    EVALUATION METRICS

**Fréchet Inception Distance (FID)** (Heusel et al., 2017) reflects the quality of the generated image. FID measures similarity of features between two image sets and the features extracted by the pre-trained Inception-V3 (Szegedy et al., 2016).

**Inception Score (IS)** Salimans et al. (2016) uses a pre-trained Inception-V3 (Szegedy et al., 2016) to classify the generated images, reflecting the diversity and quality of the images. When calculating the

Table 7: Hyperparameters for pre-training DODA. DODA leverages latent diffusion (LDM) (Rombach et al., 2022) as the base diffusion model, which uses variational autoencoder (VAE) (Kingma & Welling, 2013) to encode the image into the latent space and thus reduces the computation, so the pre-training of DODA is divided into two stages: the VAE and LDM.

| | | VAE | LDM |
|---|---|---|---|
| Dataset | | All images in GWHD | All images in GWHD |
| Target Image Shape | | $256 \times 256 \times 3$ | $256 \times 256 \times 3$ |
| Domain Reference Image Shape | | - | $224 \times 224 \times 3$ |
| Data Augmentation | Target Image | Random Rotation Random Crop Random Flip | Random Rotation Random Crop Random Flip |
| | Reference Image | - | Random Crop |
| f | | 4 | 4 |
| Channels | | 128 | 224 |
| Channel Multiplier | | 1,2,4 | 1,2,4 |
| Attention Resolutions | | - | 2,4 |
| Number of Heads | | - | 8 |
| Learning Rate | | 2.5e-6 | 2.5e-5 |
| Iterations | | 480k | 600k |
| Batch Size | | 8 | 16 |
| GPU Memory usage | | 32 GB | 16 GB |
| Computational consumption | | 20 v100-days | 14 v100-days |

Table 8: Hyperparameters for layout-to-image.

| Dataset | | COCO 2017 training | COCO 2017 training | GWHD training |
|---|---|---|---|---|
| Target/Layout Image Shape | | $256 \times 256 \times 3$ | $512 \times 512 \times 3$ | $256 \times 256 \times 3$ |
| Domain Reference Image Shape | | - | - | $224 \times 224 \times 3$ |
| Data Augmentation | Target Image | Random Flip | Random Flip | Random Rotation Random Crop Random Flip |
| | Reference Image | - | - | Random Crop |
| Base Model | | SD1.5 | COCO 256 | LDM in Table 7 |
| f | | 8 | 8 | 4 |
| Channels | | 320 | 320 | 224 |
| Channel Multiplier | | 1,2,4,4 | 1,2,4,4 | 1,2,4 |
| Attention Resolutions | | 1,2,4 | 1,2,4 | 2,4 |
| Number of Heads | | 8 | 8 | 8 |
| Learning Rate | | 2.5e-5 | 2.5e-5 | 1e-5 |
| Iterations | | 100K | 30K | 80K |
| Batch Size | | 16 | 8 | 16 |
| GPU Memory usage | | 27 GB | 25 GB | 20 GB |
| Computational consumption | | 40 v100-hours | 56 v100-hours | 40 v100-hours |

IS for Table 2, as in the original paper, we divided the data into 10 splits. The error bar for IS is the standard deviation between the splits.

**COCO Metrics** refers to fine-tuning detectors using synthetic data, and then calculating AP according to the official COCO.

**YOLO Score** uses a pre-trained YOLOX-L (Ge et al., 2021) to detect the generated image, and calculates the AP between the detection result and the input label, which reflects the ability of the generated model to control the layout.

**Feature Similarity (FS)**. As discussed in Sec.3.3, domain shift manifests in feature differences, the domain encoder should guide diffusion to generate images aligned with reference images' features. Here we use DINO-V2 (Oquab et al., 2023) to extract features from the generated images and their corresponding reference images, calculate the cosine similarity for each pair, and then compute the average similarity across multiple image pairs. Compared with FID, FS provides more fine-grained information.

# D  SYNTHETIC DATA FOR DAY-TO-NIGHT DOMAIN ADAPTATION

We also evaluated our method on a non-agricultural object detection dataset. Following Kennerley et al. (2023) and Zhang et al. (2024), we divided the BDD dataset (Yu et al., 2020) into 'daytime' and 'night', with 'daytime' as the source domain and 'night' as the target domain. The results are shown in Table 1. Although our method is designed for agricultural datasets, it can be applied to other domains as well. However, there is still a performance gap compared to SOTA methods in this task. To achieve optimal results, further improvements are needed. For example, when the target domain is known and limited, a learnable domain embedding could be used to replace our domain encoder.

Table 9: Results of Day-to-Night domain adaption on the BDD dataset. From left to right, the full class name are Pedestrian, Rider, Car, Truck, Bus, Motorcycle, Bicycle, Traffic light, Traffic sign.

| Model | Method | Bic. | Bus | Car | Rid. | Tru. | Mot. | Ped. | T-Light | T-Sign | AP |
|---|---|---|---|---|---|---|---|---|---|---|---|
| Faster RCNN | Source | 39.5 | 47.5 | 66.6 | 28.9 | 47.8 | 32.8 | 50.0 | 41.0 | 56.5 | 41.1 |
| (Ren et al., 2016) | TDD (He et al., 2022b) | 25.9 | 35.6 | 68.4 | 20.7 | 33.3 | 16.5 | 43.1 | 43.1 | 59.5 | 34.6 |
| | UMT (Deng et al., 2021) | 40.2 | 46.3 | 46.8 | 26.1 | 44.0 | 28.2 | 46.5 | 31.6 | 52.7 | 36.2 |
| | AT (Li et al., 2022) | 42.7 | 52.1 | 60.8 | 30.4 | 48.9 | 34.5 | 42.3 | 29.1 | 43.9 | 38.5 |
| | 2PCNet (Kennerley et al., 2023) | 44.5 | 55.2 | 73.1 | 30.8 | 53.8 | 37.5 | 54.4 | 49.4 | 65.2 | 46.4 |
| | ISP-T (Zhang et al., 2024) | 48.1 | 55.9 | 72.9 | 39.4 | 54.6 | 43.8 | 57.8 | 49.6 | 66.3 | 48.8 |
| YOLO X | Source | 36.4 | 45.2 | 67.5 | 26.0 | 50.6 | 24.2 | 37.8 | 56.4 | 48.8 | 39.3 |
| | Ours | 44.7 | 51.5 | 68.8 | 32.4 | 53.8 | 32.6 | 37.4 | 56.9 | 51.4 | 43.0 |

# E  MORE ABLATION

**Adapting different detectors.** To verify the generalizability of the generated data, we evaluate several detectors with various architectures and sizes. In addition to YOLOX-L, we evaluate Deformable DETR (Zhu et al., 2020), YOLOV7-X (Wang et al., 2023a), and FCOS X101 (Tian et al., 2019). As shown in Table 10, when trained solely on the official GWHD training set, the detectors struggle to recognize the "Terraref" domain, but after fine-tuning with domain-specific data generated by DODA, these models show consistent improvement (+20.1∼26.8 $AP_{50}$) in recognizing "Terraref" domain.

**Channel coding.** In this work, we use different color channels to help the model better distinguish overlapped instances and thus more accurately control the layout. We further train a DODA without channel coding, which results in a lower YOLO Score and higher FID, as shown in Table 11.

**Reference image selection method.** In the main experiments, we randomly sample reference images $x_{ref}$ from the entire set of target domain images $x$. In this section, we investigate how the choice of reference images affects the generated data. We first sample different numbers of images from $x$ to create reference pools $x_{pool}^i$ of varying sizes, and then randomly sample $x_{ref}$ from each $x_{pool}^i$. For each $x_{pool}^i$, we repeat the sampling process 5 times to compute the standard deviation. As shown in Table 12, when the reference pool is extremely small, the diversity of $x_{ref}$ is low, resulting in low AP

scores, and the standard deviation is large because the sampling bias is amplified. Once the size of the $x_{pool}^i$ exceeds 100, the AP stabilizes.

**Number of generated images.** We investigate changes in the performance of using different amounts of generated data. As shown in Fig. 3, for most domains, a dataset consisting of 200 synthetic images is sufficient to convey the characteristics of the target domain. Increasing the number of images does not significantly improve performance. To ensure consistency across experiments, we use 200 images by default for all domains.

Table 10: Effectiveness of DODA on different detectors. All detectors are trained on the GHWD training set, and their $AP_{50}$ in the "Terraref" domain is reported as the baseline. After fine-tuning with DODA-generated data, detectors of different sizes and architectures show consistent improvement.

| Method | Params | w/o DODA | w/ DODA |
|---|---|---|---|
| Deformable DETR | 41M | 10.4 | 37.2(+26.8) |
| YOLOX L | 54M | 30.5 | 50.7(+20.2) |
| YOLOV7 X | 71M | 27.2 | 47.3(+20.1) |
| FCOS X101 | 90M | 20.1 | 41.5(+21.4) |

Table 11: Ablations on the layout channel coding. Channel coding can help the model more accurately control layout.

| Channel coding | YOLO Score↑ | | | | | |
|---|---|---|---|---|---|---|
| | mAP | $AP_{50}$ | $AP_{75}$ | $AP^s$ | $AP^m$ | $AP^l$ |
| ✗ | 26.4 | 67.8 | 14.5 | 20.0 | 31.3 | 28.6 |
| ✓ | 27.4 | 70.0 | 15.3 | 20.8 | 32.7 | 29.9 |

Table 12: Ablations on the selecting of reference images.

| References pool size | $AP_{50}$ |
|---|---|
| 0 | 30.5 |
| 10 | 37.22±8.89 |
| 50 | 45.44±3.64 |
| 100 | 48.00±1.92 |
| 200 | 47.92±1.91 |
| 400 | 49.36±1.55 |
| 800 | 47.48±1.98 |
| 1600 | 48.58±1.95 |

## F  FAILURE CASE ANALYSIS

On the "Ukyoto_1" domain, the improvement in mAP is minimal. As shown in Fig. 4 left, we observed many images with an unusual black area. Compared to images without black edges (Fig. 4 middle), these images also show poorer alignment with the given layout. The black area and poorer alignment negatively affect the quality of generated data. Upon further inspection, we found that these black regions originate from the real images used for pre-training (as illustrated in Fig. 4 right). When preparing the pre-training dataset, it may be necessary to clean up such images to improve data quality.

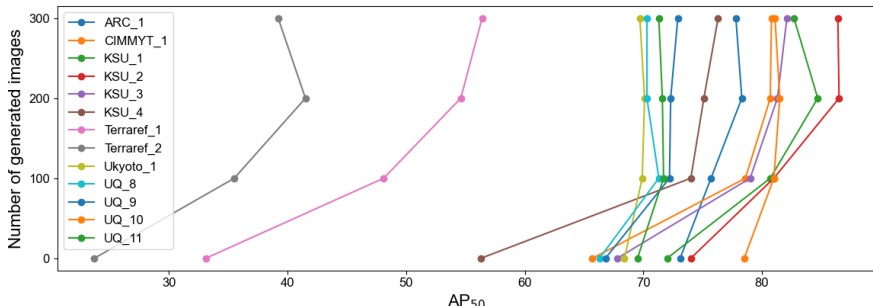

Figure 3: Ablations on the number of generated images. For most domains, 200 generated images are sufficient.

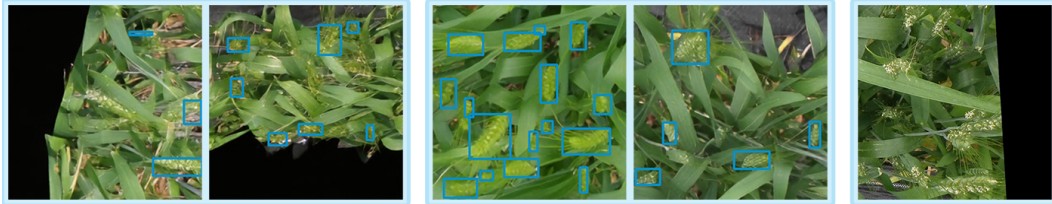

Figure 4: Examples of generated images in domain "Ukyoto_1". Left, many generated images have strange black edges. Middle, normal generated images, which are better aligned with the input layout (blue bounding boxes) than the images with black edges. Right, some real images used for pre-training also have black edges.

# G   QUALITATIVE COMPARISONS WITH PREVIOUS L2I METHODS ON COCO

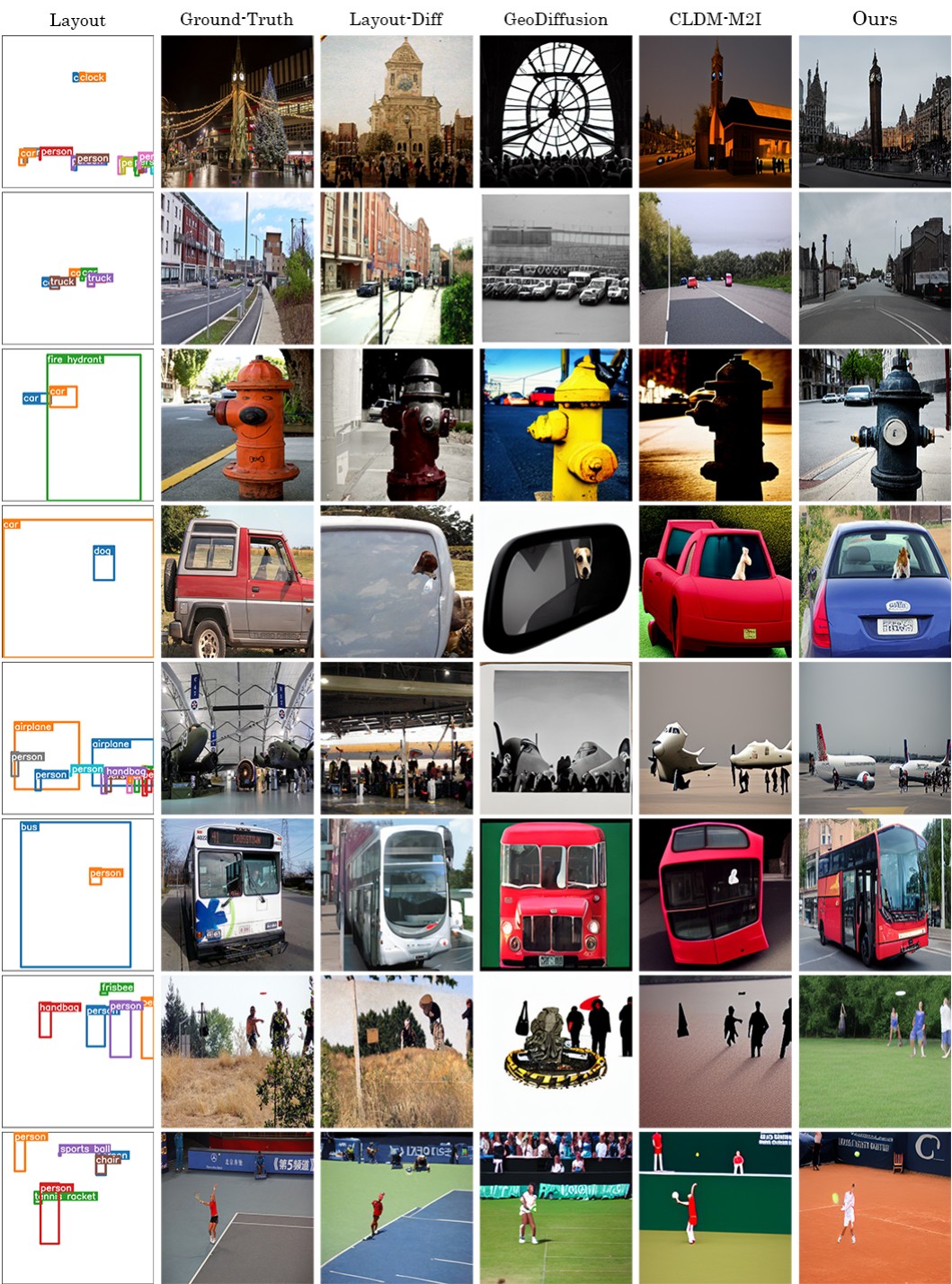

Figure 5: Visualization of comparisons between our proposed LI2I method and previous LT2I methods on COCO. LI2I generates images with more detail and greater control over layout, especially for small objects.

