# OpenReview forum: "DODA: Diffusion for Object-detection Domain Adaptation in Agriculture"
_ICLR.cc/2025/Conference — Submitted to ICLR 2025_

### Official Review · Reviewer_r2nh · 2024-10-27

**Soundness:** 3
**Presentation:** 3
**Contribution:** 3
**Rating:** 5
**Confidence:** 5

**Summary:**

This paper presents DODA, a framework that uses diffusion models to improve object detection in diverse agricultural domains. By integrating domain embeddings and a layout-image-to-image (LI2I) method, DODA generates domain-specific images without extra training, enhancing model adaptability across different environments. Tests on the Global Wheat Head Detection dataset show substantial AP improvements, underscoring DODA's potential for accessible, customized object detection in agriculture.

**Strengths:**

1.  DODA incorporates external domain embeddings and is able to generate high-quality detection data for new domains without additional training. This approach is inspiring.
2.  Significant improvements are witnessed both in agriculture detection and L2I generation performance.

**Weaknesses:**

1.	Fields of application. Restricting the research to the agriculture domain limits the impact of this paper. I am curious if it could also be applied to other fields, such as autonomous driving."
2.	The pipeline of ‘layout-text-to-image’ can be seen as a simplified and derived version of ControlNet [1] , lacking innovation in this aspect.
3.	In Figure 2(c), most of the content is dedicated to describing existing text-based L2I methods, while the description of the proposed method is limited. Additionally, the figure only includes content related to 'Encoding Layout Images' (Section 3.4), lacking description of 'Incorporating Domain Embedding' (Section 3.3).

[1] Adding Conditional Control to Text-to-Image Diffusion Models, ICCV2023

**Questions:**

1.	As a L2I generation method, I have questions about the performance of DODA. The pipeline of ‘layout-text-to-image’ can be seen as a simplified and derived version of ControlNet [1]. However, as GeoDiffusion [2] reported, ControlNet can only achieve 25.2 mAP in COCO 2017 valuation set as a L2I generation method by converting layouts to masks, while DODA claims 42.5 mAP which has an improvement of 17.3%. Therefore, I am curious about the reasons behind huge performance improvement.
2.	Fields of application. See Weaknesses 2.

[1] Adding Conditional Control to Text-to-Image Diffusion Models, ICCV2023
[2] Geodiffusion: Text-prompted geometric control for object detection data generation, ICLR2024

---

> ### Author Response · Authors · 2024-11-22
> **Response to Reviewer r2nh**
>
> Thank you for your review and the insightful comments.
>
> **W1 & Q2 Additional results and baseline methods on BDD day-to-night task.**
>
> Please refer to the General Response.
>
> ---
>
> **W2 Additional baseline method on COCO dataset.**
>
> On the COCO dataset, we additionally evaluate the performance of ControlNet [1] with mask as condition, following the ControlNet paper. The results are provided in Table R2. We have also added these results to Table 2 in the main paper and included images generated by ControlNet in Figure 5.
>
> **Table R2. Comparison on COCO.**
>
> | Method | mAP | AP50 | FID | IS |
> |:-:|:-:|:-:|:-:|:-:|
> | GeoDiffusion | 27.7 | 40.7 | 28.8 | 26.4 |
> | ControlNet M2I | 39.5 | 50.3 | 46.9 | 20.2 |
> | **Ours** | **42.5** | **56.1** | **24.9** | **29.4** |
>
> As shown in Table R2, our LI2I method not only provides better layout control, but also significantly outperforms ControlNet in terms of image quality (FID) and image diversity (IS), which is supported by the visual results in Figure 5. The images generated by ControlNet closely follow the mask contours, leading to distorted object appearances and simplistic backgrounds. The method of using mask as a condition in ControlNet is too strict for image generation.
>
> **Advantages of LI2I over ControlNet in Data Generation:**
> 1. **Ease of Layout Generation**: A layout image can be easily created by randomly generating the coordinates of bounding boxes, whereas masks are hard to obtain programmatically. This will be the primary block for generating random data.
> 2. **Better Image Quality**: Compared to semantic masks used in ControlNet, our layout image has more moderate constraints on objects, allowing for the generation of more accurate and diverse objects, and the overall layout is also more accurate. These factors are vital for high-quality synthetic data.
>
> Thus, our LI2I is crucial for generating high-quality synthetic detection data.
>
> ---
>
> **W3 Reorganization of Figure 2(c)**
>
> Thank you for your valuable feedback. We have updated Figure 2(c) to focus on the model architecture of DODA rather than comparing the two L2I methods. The revised figure now includes both the domain encoding described in Section 3.3 and the layout encoding covered in Section 3.4.
>
> ---
>
> **Q1 Re-implementation of ControlNet**
>
> We re-implement the Mask-to-Image method of ControlNet [1]. To ensure a fair comparison, we keep the basic settings consistent with those used in our LI2I method. As shown in Table R2, in our implementation, ControlNet achieves a higher mAP compared to GeoDiffusion[2]. The authors of GeoDiffusion mentioned that they used the 'binary semantic mask' as the condition when implementing ControlNet [3], and we assume this difference in implementation that leads to the performance difference.
>
> [1] Adding Conditional Control to Text-to-Image Diffusion Models. ICCV2023
>
> [2] GeoDiffusion: Text-Prompted Geometric Control for Object Detection Data Generation. ICLR2024
>
> [3] GeoDiffusion: Text-Prompted Geometric Control for Object Detection Data Generation. openreview, Response to Reviewer VRFf, Q1 (1)

---

> > ### Comment · Reviewer_r2nh · 2024-11-25
> >
> > Thanks for your effort.
> >
> > I understand the performance of ControlNet M2I comes from additional masks, which are not used in your LI2I pipeline. However, why does such a direct and straightforward 'layout-text-to-image' paradigm achieve such high performance, even surpassing methods that utilize additional masks? From my point of view, the LI2I method merely adopts a vision encoder to convert layout images.

---

> > > ### Author Response · Authors · 2024-11-25
> > > **Response to Reviewer r2nh**
> > >
> > > I sincerely appreciate your feedback!
> > >
> > > When using masks as conditions, the model is forced to generate objects that conform to mask contours. This strict pixel-level constraint may conflict with the denoising process, leading to deviations in the appearance of objects. In some cases, we can even observe that masked regions become completely black, failing to generate meaningful objects (as shown in the second-to-last image generated by ControlNet in Figure 5). Since the YOLO Score is calculated through a detector, when the detector fails to recognize certain objects, the YOLO Score decreases.
> > >
> > > Our approach uses layout images as conditions. During model training, the model only has access to objects' bounding boxes. After convergence, it develops the ability to transform bounding boxes into objects. This weaker constraint allows the model to adjust object shapes within the bounding boxes, resulting in more natural-looking objects. Consequently, this leads to higher image quality and better object recognition by detectors, so our method performs better than ControlNet M2I overall.

---

> > > ### Author Response · Authors · 2024-11-29
> > > **Follow up on rebuttal feedback**
> > >
> > > As we near the conclusion of the discussion phase, we would like to follow up on our responses to your valuable feedback. We understand that your schedule may be demanding, but we would be grateful for any additional feedback on our responses or confirmation that our revisions have adequately addressed your concerns. If there are any remaining questions, we are more than happy to clarify them.
> > >
> > > Thank you once again for your constructive and thoughtful review.

---

### Official Review · Reviewer_5rAY · 2024-11-01

**Soundness:** 1
**Presentation:** 2
**Contribution:** 2
**Rating:** 3
**Confidence:** 5

**Summary:**

The authors propose a diffusion model framework for generating detection data from a range of agricultural scenarios.  Performance is tested on the Global Wheat Head Detection dataset.

**Strengths:**

Incorporation of this approach appears to improve detection results, often significantly.

The ablation studies explore pre-training dataset size, encoder impact, number of layers, and embedding positioning.

**Weaknesses:**

Only a single dataset is explored.  Given the claim that this is a unified framework for domain adaptation, performance should be tested over a wide range of datasets, crops, and environments.  Having a single dataset significantly limits the claims which can be made around applicability and general usefulness.

Additionally, there is nothing in the method that seems to be specifically agricultural-focused.  Is this approach necessary for agricultural setting?  Would it work in other domains?

The writing (both at the sentence/grammatical level as well as high-level sentence organization) could be improved in places- some of the clauses are dangling and incomplete.

Related works around object detection in the agricultural space are largely dominated by a narrow group of authors.  There is significant amount of work done in the detection/counting space across many crops- I would suggest the authors look at the work done by these labs including works such as Ghosal 2019, Malambo 2019, Osco 2020, Gene-Mola 2020, Akiva 2020, Akiva 2021, Hobbs 2021, and others.  It is important to represent the broad work of this area across crops of interest done by a wide range of labs.

**Questions:**

The language in the Related Work is particularly unclear.  The sentences tend to be fragmented and do not follow well.  I would suggest a rather substantial revision of this section.

The last sentence of the first paragraph of 3.3 is extremely length and unclear and should be broken up.  The first three paragraphs of 3.3 are largely narrative as opposed to method dominant- I would suggest putting this motivating information in the introduction and keep only the description of the present method in this section.

Figure 2b is very unclear- is the reader supposed to interpret that as 6 subpanels that have been snacked across two rows?  The caption doesn't make this clear either.

See weaknesses.  It's unclear why this method is directed at a single domain and not domain adaptation broadly.

It would be helpful to show Table 3 in terms of number of samples, not percent, so the reader knows the actual magnitude of data.

---

> ### Author Response · Authors · 2024-11-22
> **Response to Reviewer 5rAY**
>
> Thank you for your review and the valuable suggestions.
>
> **W1 & Q4 Additional experiments on domain adaption.**
>
> The GWHD dataset consists of 47 sub-datasets collected from multiple countries, covering a wide range of environments, crop varieties, growth stages, and imaging pipelines. As demonstrated in Table 1 of our paper, our method consistently improves performance across multiple domains. We believe that the results reflect the robustness of our method under varying conditions.
>
> To address your concern, we conducted additional experiments on a new wheat dataset beyond GWHD. While GWHD images were collected from ground-level, the new dataset consists of images captured by UAV, introducing significant spatial resolution differences. As shown in the first row of Table R3, our method effectively helps the detector adapt to such variations.
>
> **Table R3. Domain adaption on other agricultural datasets.**
>
> | Crop | AP | AP50 | Placform |
> |:-:|:-:|:-:|:-:|
> | wheat | 13.3 | 35.5 | UAV |
> | +Ours | 28.2(+14.9) | 54.9(+19.4) | |
> | | | | | |
> | Sorghum | 17.3 | 40.0 | 12.6 | UAV |
> | +Ours | 29.4(+12.1) | 70.5(+30.5) | |
>
> Furthermore, we tested our approach on sorghum as the additional crop. The second row of Table R3 demonstrates that our method can achieve cross-crop adaptation.
>
> As highlighted in Section 4.3, diverse images can provide better adaptation performance. However, to the best of our knowledge, there are no object detection datasets for other crops that have been built in a worldwide collaboration like GWHD. In addition, it is common in this field for researchers to collect private datasets and train private models. We hope that this paper will encourage the community to collaborate on building more diverse datasets.
>
> ---
>
> **W2 & Q4 Additional results and baseline methods on BDD day-to-night task.**
>
> Please refer to the General Response.
>
> ---
>
> ***W3 & Q1 Refinement of Writing for Clarity and Flow***
>
> Upon careful review by a native English-speaking co-author, many less-than-fluent sentences were rewritten. Several improvements have also been made to sentence organization within a paragraph, as well as paragraph organization itself, particularly in the "Related Works" section.
>
> ---
>
> **W4 New section in Related Work for broader context of our work**
>
> We have added a new section in the Related Work titled "Detection and Counting in Agriculture," where we discuss the suggested papers and other relevant works. This addition provides readers with a clearer understanding of the background. We hope this enhancement addresses your concern.
>
> ---
>
> **Q2 Revision of Section 3.3**
>
> We have revised the first paragraph of Section 3.3 for clarity and conciseness, and split the last sentence to improve readability.
> Regarding the first three paragraphs of Section 3.3, we intended them to explain the rationale behind the design of the method. We consider that moving all of this content from Section 3.3 would disrupt the flow of the section. However, we have moved most of the discussion on domain adaptation to the Introduction and Related Work sections. We also minimized the description of the three paragraphs to keep the focus on the method itself.
>
> ---
>
> **Q3 Reorganization of Figure 2(b).**
>
> Thank you for your feedback. The six sub-images in Figure 2(b) are from different layers of the U-Net architecture. The top row represents features from the encoder, and the bottom row represents features from the decoder. To make this clearer, we have resized the images and added arrows between them. In addition, we have updated the figure caption to explicitly state that these sub-images are from different U-Net layers for clarity.
>
> ---
>
> **Q5 Suggestion on Table 3**
>
> Following your suggestion, we have updated Table 3 to display the actual number of unlabeled images instead of percentages, making the table more informative for readers.

---

> ### Author Response · Authors · 2024-11-29
> **Follow up on rebuttal feedback**
>
> As we near the end of the discussion phase, we wanted to follow up on the responses to your comments. We have carefully addressed each point raised and hope our responses demonstrate the merits of our paper. We appreciate the time and effort you have dedicated to reviewing our work, and we would be grateful for any additional feedback on our responses or confirmation that the responses have addressed your concerns.

---

### Official Review · Reviewer_XiWb · 2024-11-07

**Soundness:** 3
**Presentation:** 2
**Contribution:** 3
**Rating:** 6
**Confidence:** 4

**Summary:**

This paper presents DODA, an new framework utilizing diffusion models to improve object detection across diverse agricultural environments. DODA tackles the challenge of adapting detection models to various agricultural domains by integrating external domain embeddings alongside a layout-to-image (LI2I) method, enabling the generation of high-quality, domain-specific detection data without the need for additional model retraining. Experimental evaluations on the Global Wheat Head Detection dataset reveal substantial enhancements in detection performance across multiple domains, with a notable average increase in accuracy.

**Strengths:**

1. the paper introduces a novel way to achieve few shot domain adaptation by using diffusion model, leading to more accurate and contextually relevant synthetic data.

2. The research focus on agriculture data is important and will benefit the community.

3. The DODA framework is designed to be user-friendly, requiring minimal training.

**Weaknesses:**

1.  In the related work section, the authors are suggested to enrich the related works in few shot domain adaptation task. According to the description in Section 4.2, the adaptation relies on few samples from the new domain, which is within the few shot domain adaptation setting. However the authors did not clearly point it out while renaming it as DODA.


2. The authors should enrich the details regarding how to choose the domain reference images. Is it randomly? How about the variance of the performance on different set of reference images?


3. which visual encoder is used as mentioned on line 210? Can it generalize to other visual encoder?


4. The authors are suggested to compare the proposed approach with other existing few shot domain adaptation methods to demonstrate the effectiveness of the proposed new method, e.g., a,b,c,d,e, and f.

a. Jing, T., Xia, H., Hamm, J., & Ding, Z. (2023). Marginalized augmented few-shot domain adaptation. IEEE Transactions on Neural Networks and Learning Systems.

b. Wu, Y., Li, Z., Wang, C., Zheng, H., Zhao, S., Li, B., & Tao, D. (2024). Domain re-modulation for few-shot generative domain adaptation. Advances in Neural Information Processing Systems, 36.

c..Gao, Y., Yang, L., Huang, Y., Xie, S., Li, S., & Zheng, W. S. (2022, October). Acrofod: An adaptive method for cross-domain few-shot object detection. In European Conference on Computer Vision (pp. 673-690). Cham: Springer Nature Switzerland.

d. Gao, Y., Lin, K. Y., Yan, J., Wang, Y., & Zheng, W. S. (2023). AsyFOD: An asymmetric adaptation paradigm for few-shot domain adaptive object detection. In Proceedings of the IEEE/CVF Conference on Computer Vision and Pattern Recognition (pp. 3261-3271).

e. Nakamura, Y., Ishii, Y., Maruyama, Y., & Yamashita, T. (2022). Few-shot adaptive object detection with cross-domain cutmix. In Proceedings of the Asian Conference on Computer Vision (pp. 1350-1367).

f. Corral-Soto, E. R., Nabatchian, A., Gerdzhev, M., & Bingbing, L. (2021, May). Lidar few-shot domain adaptation via integrated cyclegan and 3d object detector with joint learning delay. In 2021 IEEE international conference on robotics and automation (ICRA) (pp. 13099-13105). IEEE.


5. On line 406, why did the authors choose 200 as the generated image number? Is there any ablation study regarding this number?


6. Lack of failure case analysis.

**Questions:**

1. Could the authors enrich the related works in few-shot domain adaptation? Given the reliance on few samples from new domains in Section 4.2, would it be appropriate to explicitly situate this work within the few-shot domain adaptation setting?

2. How are domain reference images selected for DODA? Is the selection random, and has the variance in performance across different sets of reference images been analyzed?

3. Which visual encoder is specifically used in line 210, and can the approach generalize to other types of visual encoders?

4. To strengthen the validation of DODA, would the authors consider comparing it against other few-shot domain adaptation methods, such as those proposed by Jing et al. (2023), Wu et al. (2024), Gao et al. (2022), Gao et al. (2023), Nakamura et al. (2022), and Corral-Soto et al. (2021)?

5. In line 406, why was the number of generated images set to 200? Was any ablation study conducted to examine the impact of this number?

6. Could the authors provide a failure case analysis to highlight scenarios where DODA may struggle, as this would offer insights into potential limitations of the approach?

---

> ### Author Response · Authors · 2024-11-22
> **Response to Reviewer XiWb**
>
> Thank you very much for your review and the inspiring comments.
>
> **W1 & Q1 Enrich the related works in few-shot domain adaptation**
>
> Our approach consists of two main parts: (1) generating images for the target domain, and (2) adapting the detector to the target domain. For the image generation task, we do need reference images from the target domain, and this subtask is a form of few-shot generative domain adaptation. However, our primary goal is to adapt the detector using the generated images. While few-shot detection domain adaptation typically requires a small number of annotations from the target domain, our approach is designed to eliminate the need for manual annotations, thereby making object detection more accessible. Therefore, we consider our method to be closer to unsupervised domain adaptation overall.
>
> We appreciate your suggestion and have expanded our discussion of few-shot domain adaptation in the Related Work section to provide readers a better context for our method.
>
> ---
>
> **W2 & Q2 Ablation study of the reference images selection**
>
> In the main experiments, we randomly sample reference images $x_{ref}$ from the entire set of target domain images $x$.
> Following your suggestion, we also report how the choice of reference images affects the generated data in Appendix E. We first sample different numbers of images from $x$ to create reference pools $x_{pool}^i$ of varying sizes, and then randomly sample $x_{ref}$ from each $x_{pool}^i$. For each $x_{pool}^i$, we repeat the sampling process 5 times to compute the standard deviation.
>
> **Table R4. Ablations on the selecting of reference images.**
> | References pool size | AP50 |
> |:-:|:-:|
> | 10 | 37.22±8.89 |
> | 50 | 45.44±3.64 |
> | 100 | 48.0±1.92 |
> | 200 | 47.92±1.91 |
> | 400 | 49.36±1.55 |
> | 800 | 47.48±1.98 |
> | 1600 | 48.58±1.95 |
>
> As shown in Table R4, when the reference pool is extremely small, the diversity of $x_{ref}$ is low, resulting in low AP scores, and the standard deviation is large because the sampling bias is amplified. Once the size of the $x_{pool}^i$ exceeds 100, the AP stabilizes.
>
> ---
>
> **W3 & Q3 Clarification on the choice of domain encoder**
>
> The choice of the domain encoder is flexible. As shown in Table 4 of main paper, pretrained vision encoders such as MAE [1] and ResNet [2] can serve as the domain encoder, so we didn't specify a particular encoder in line 210. Our experiments showed that using the vision encoder of CLIP [3] gave the best results, and we use it by default. We have added an clear reference to Table 4 in Section 3.3, so that readers can easily access the ablation study on the choice of the domain encoder.
>
> [1] Masked autoencoders are scalable vision learners. CVPR2022
> [2] Deep residual learning for image recognition. CVPR2016
> [3] Learning transferable visual models from natural language supervision. ICML2021
>
> ---
>
> **W4 & Q4 Additional results and baseline methods on BDD day-to-night task.**
>
> Please refer to the General Response.
>
> ---
>
> **Q5 Ablation study on the number of generated images**
>
> In Appendix E, we add an experiment to clarify the effect of the synthetic dataset size on the fine-tuning effect. As illustrated in Figure 3, for most domains, a dataset consisting of 200 synthetic images is sufficient to convey the information of the target domain. Further increases in the number of images do not have a significant impact on performance. To ensure consistency across experiments, we use 200 images by default for all domains.
>
> ---
>
> **Q6 Failure case analysis**
>
> We have added a new section titled "Failure Case Analysis" as Appendix F. Specifically, we analyzed the minimal improvement on the ``Ukyoto_1'' domain. As highlighted in this section, we found that some of the generated images contain strange black areas and misalignment with the input layout, which negatively impacted the quality of the generated data.
>
> Further investigation revealed that these black edges actually originated from the real images used during the pre-training. These images, with unnatural characteristics, affect the quality of the generated data. Therefore, it's important to filter out such images when collecting the pre-training dataset. Thank you for the inspiring suggestion!

---

> > ### Comment · Reviewer_XiWb · 2024-11-25
> > **Further questions**
> >
> > Thank you for the detailed comments.
> >
> > I still have question regarding my W1 and W4 concerns. The authors claimed that their task setting is in few-shot generative domain adaptation. However using the generated samples, existing few-shot domain adaptation approaches can also be leveraged to achieve the domain adaptation in your task.
> >
> >  In that case I think those approaches can be used in your own task to enrich the benchmark and comparison, but not on BDD dataset (I did not mention it in my review).
> >
> > I would appreciate if the authors can address this concern since I think the  experiment part is weak to prove the effectiveness of the proposed method.
> >
> > Thank you.

---

> > > ### Author Response · Authors · 2024-11-26
> > > **Response to Further questions**
> > >
> > > Thank you for your suggestion.
> > >
> > > For unsupervised object detection domain adaptation (UDA), the accessible datasets are $D_s = (x_s^n, y_s^n)$  and $D_t = (x_t^m)$  where $D_s$ is the source domain，$x_s^n$  are the images from $D_s$ and $y_s^n$ are the corresponding bounding box annotations, $D_t$  is the target domain, with $x_t^m$ as its images. For few-shot domain object detection adaptation (FSDA), the accessible datasets are $D_s =(x_s^n, y_s^n)$ and $D_t = (x_t^m, y_t^m)$.
> > >
> > > Thus, the main difference between FSDA and UDA is, FSDA requires access to the target domain labels $y_t^m$, whereas UDA does not. Our method only utilizes the target domain images $x_t^m$ without requiring the $y_t^m$, making it closer to the UDA setting.
> > >
> > > Following your suggestion, we have added comparisons with few-shot methods on the 'Terraref' domain. The results are shown in Table R5.
> > >
> > > **Table R5. Comparisons with few-shot domain adaptation methods.**
> > >
> > > | Method | Use source data | target data | AP | AP50 |
> > > |:-:|:-:|:-:|:-:|:-:|
> > > | Baseline | √ | - | 14.5 | 38.6 |
> > > | Fine-tuning | × | Generated | 19.4 | 53.6 |
> > > | | | | | |
> > > | AcroFOD [1] | √ | Real | 19.2 | 50.5 |
> > > | AcroFOD [1] | √ | Generated | 19.5 | 51.1 |
> > > | | | | | |
> > > | AsyFOD [2] | √ | Real | 17.8 | 47.0 |
> > > | AsyFOD [2] | √ | Generated | 19.4 | 52.3 |
> > >
> > > Simply fine-tuning the detector using the generated data outperforms FSDA methods that using real data.
> > >
> > > Using our generated data for FSDA can improve the performance of these FSDA methods. However, these FSDA methods require training the detector from scratch using both the source domain data and the generated data. This significantly increases the computational cost, with training time for one epoch rising by 150 times (200 generated images vs. 200 generated images + 30,000 source images).
> > >
> > >
> > > **Reference**
> > >
> > > [1] AcroFOD: An Adaptive Method for Cross-domain Few-shot Object Detection. ECCV2022
> > >
> > > [2] AsyFOD: An Asymmetric Adaptation Paradigm for Few-Shot Domain Adaptive Object Detection. CVPR2023

---

> ### Comment · Reviewer_XiWb · 2024-11-27
> **Another question**
>
> Thanks the authors for the response.
>
> You have proved that your image generation method also works in two FSDA approaches which is good.
>
> Then how do the image generation approaches (e.g., LayoutDiffusion) listed in Table II work on Table I? I would like to see if your image generation pipeline works better than other image generation pipelines for this task, not in the way as evaluated by FID in Table II but in the performance of object detection under domain adaptation setting.
>
> In that case it would be an evaluation regarding how large the quality of the generated images will affect the performance for the domain adaptation. If your method still shows better performance compared with other image generation approaches in that way, I will consider to increase my score.

---

> > ### Author Response · Authors · 2024-11-28
> > **Response to Further Question**
> >
> > We greatly appreciate your valuable feedback and comments, which help us to improve our work.
> >
> > We selected two top-performing Layout-to-Image (L2I) methods in Table 2, GeoDiffusion [1] and ControlNet Mask2Image (M2I) [2], for comparison with our approach.
> >
> > For GeoDiffusion, we fine-tuned the official weights on the GWHD training set for 60 epochs. Due to its text-based layout encoding, it’s incompatible with our domain encoder. Therefore, we use DDIM inversion to enable it to synthesize images close to the style of the target domain (img2img could not generate meaningful images).
> >
> > ControlNet M2I is compatible with our domain encoder, so we use our pre-trained latent diffusion directly. We utilize SAM to convert bounding boxes into masks as the condition for ControlNet M2I. When generating data, ControlNet M2I requires masks, which are difficult to obtain programmatically, so we directly use the mask of ground truth.
> >
> > As shown in Table R6, our method demonstrates the most significant performance improvement in the 'Terraref' domain.
> >
> > | Method | AP | AP50 |
> > |:-:|:-:|:-:|
> > | Baseline | 10.7 | 30.5 |
> > | GeoDiffusion | 15.1 | 40.2 |
> > | ControlNet M2I | 18.8 | 48.8 |
> > | DODA (ours) | 21.1 | 51.3 |
> >
> > **Reference**
> >
> > [1] GeoDiffusion: Text-Prompted Geometric Control for Object Detection Data Generation. ICLR2024
> >
> > [2] Adding Conditional Control to Text-to-Image Diffusion Models. ICCV2023

---

> > > ### Comment · Reviewer_XiWb · 2024-11-28
> > > **Response**
> > >
> > > Dear Authors,
> > >
> > > great work, then please modify your paper accordingly based on your responses. Thank you for the effort during response.
> > >
> > > I will increase my score to 6.
> > >
> > > Your reviewer.

---

> > > > ### Author Response · Authors · 2024-11-28
> > > > **Response to Reviewer XiWb**
> > > >
> > > > Thank you so much for your acknowledgement of this work! We have added these results to the paper and will update the pdf when possible.

---

### Official Review · Reviewer_xoHt · 2024-11-18

**Soundness:** 3
**Presentation:** 3
**Contribution:** 2
**Rating:** 6
**Confidence:** 3

**Summary:**

The paper discuss the development of the DODA framework for object detection in agriculture, showcasing its effectiveness in generating high-quality detection data across diverse agricultural domains.

**Strengths:**

1. Decouples domain-specific feature learning from the diffusion model, improving adaptability across diverse agricultural environments.
2. Introduces the LI2I method to enhance control over image layouts, leading to improved label accuracy.
3. Demonstrates the effectiveness of pre-training with additional unlabeled data in enhancing the quality of generated detection data.

**Weaknesses:**

1. The experiment appears to be somewhat limited. While the proposed method is tailored for agricultural settings, I would recommend the authors to transfer it to natural environments, such as cityscapes, to compare its effectiveness.
2. The method proposed in this paper does not seem to specifically address issues present in agricultural settings.
3. There is a lack of essential visualization of intermediate processes and comparisons.

**Questions:**

1. The number of methods selected for comparison is not sufficient and not up-to-date, it is recommended that the authors consult the latest works in the relevant field
2. The authors only provide quantitative results, which is not enough, combining qualitative results is necessary, why can work well on different domains.
3. Lack of necessary limitation analysis.

---

> ### Author Response · Authors · 2024-11-23
> **Response to Reviewer xoHt**
>
> **W1&W2 Additional results and baseline methods on BDD day-to-night task.**
>
> Please refer to the General Response.
>
> ---
>
> **W3 & Q2 Visualization of Generated Data and Detectors' Results**
>
> The left of Figure 1 shows the data we generated for different domains, and we have added a reference to it in Section 4.1.1 to help readers find this content more easily.
>
> Additionally, we hace included more visualizations of the generated data in the appendix, along with qualitative results of different detectors before and after fine-tuning,and will update the pdf when possible.
>
> ---
>
> **Q1**
>
> In Table 2, we compared our proposed LI2I method with previous SOTA Layout-to-Image methods, and our method significantly outperforms the previous approaches. In Table 10, we demonstrated the effectiveness of our method across different detectors. Since our method is independent of the detector, we selected detectors with diverse architectures and sizes as representatives, showing that our approach helps these varied detectors adapt to new domains. However, we will expand our experiments to include more detectors to further validate our method.
>
> ---
>
> **Q3 Limitation and Failure case analysis**
>
> In Appendix F, we have added a detailed analysis of failure cases, focusing on the limited AP improvements observed in the "Ukyoto_1" domain. We found that the generated data is sensitive to the real images used for pre-training. Specifically, unnatural black edges in the real images negatively impact the quality of the generated data. Therefore, careful filtering of such images is essential when constructing pre-training datasets.

---

> ### Author Response · Authors · 2024-11-29
> **Follow up on rebuttal feedback**
>
> We are grateful for the valuable feedback and comments you provide, which help us to improve our work. As we approach the end of the discussion phase, we wanted to follow up regarding your insightful comments. We have carefully and thoroughly addressed each point you raised, aiming to provide clear and comprehensive responses that highlight the contributions of our work. We understand that your schedule may be demanding, but we would be grateful for any additional feedback on our responses or confirmation that our revisions have addressed your concerns. If there are any remaining questions, we are more than happy to clarify them. Otherwise, we respectfully request that you consider adjusting your rating if you feel the revisions have strengthened the paper.
>
> Thank you once again for your thoughtful and constructive review. We look forward to your feedback.

---

### Author Response · Authors · 2024-11-22
**General Response**

We sincerely appreciate the detailed reviews, as well as the positive and constructive feedback from all the reviewers. While we have addressed each reviewer’s comments individually, we also provide a response to the common concern below:

**Can our method be directly applied to other domains such as autonomous driving?**

We evaluated our method on the Day-to-Night adaptation task using the BDD100K dataset [1]. Following 2PCnet [2] and ISP-Teacher [3], we divided the BDD dataset into 'daytime' and 'night', with 'daytime' as the source domain and 'night' as the target domain. TThe results are shown in Table R1 and have been included in Table 9 of the main paper. As Table R1 illustrates, while our method is primarily designed for agriculture, it is also applicable to the BDD dataset.

**Table R1 Results of Day-to-Night domain adaption on the BDD dataset.**

| Model | Method | Bic. | Bus | Car | Rid. | Tru. | Mot. | Ped. | T-Light | T-Sign | AP|
|:-:|:-:|:-:|:-:|:-:|:-:|:-:|:-:|:-:|:-:|:-:|:-:|
| Faster RCNN| Source | 39.5 | 47.5 | 66.6 | 28.9 | 47.8 | 32.8 | 50.0 | 41.0 | 56.5 | 41.1|
| | TDD [4] | 25.9 | 35.6 | 68.4 | 20.7 | 33.3 | 16.5 | 43.1 | 43.1 | 59.5 | 34.6|
| | UMT [5] | 40.2 | 46.3 | 46.8 | 26.1 | 44.0 | 28.2 | 46.5 | 31.6 | 52.7 | 36.2|
| | AT [6] | 42.7 | 52.1 | 60.8 | 30.4 | 48.9 | 34.5 | 42.3 | 29.1 | 43.9 | 38.5|
| | 2PCNet [2] | 44.5 | 55.2 | 73.1 | 30.8 | 53.8 | 37.5 | 54.4 | 49.4 | 65.2 | 46.4|
| | ISP-T [3] | 48.1 | 55.9 | 72.9 | 39.4 | 54.6 | 43.8 | 57.8 | 49.6 | 66.3 | 48.8|
| |  |  |  |  |  |  |  |  |  |  |  |
| YOLO X| Source | 36.4 | 45.2 | 67.5 | 26.0 | 50.6 | 24.2 | 37.8 | 56.4 | 48.8 | 39.3|
| | Ours | 44.7 | 51.5 | 68.8 | 32.4 | 53.8 | 32.6 | 37.4 | 56.9 | 51.4 | 43.0|

Our main goal is to reduce the barriers for breeders and growers to use object detectors in their personalized environments. Factors such as genetic differences, environmental variations and differences in image pipeline contribute to domain shifts. In addition, images taken by the same user at different times can differ significantly due to different weather and plant growth stages. These factors require our method to efficiently handle continuous streams of new, unseen images. To achieve this, we introduced an image encoder to extract domain embeddings that guide data generation, enabling the detector to adapt effectively to new domains (On 4090, the entire process of data generation and detector fine-tuning takes less than 3 minutes, yielding a maximum improvement of 15.6 AP and an average improvement of 7.5 AP). In agriculture, while the focus is typically on a single crop, the visual characteristics within the crop can vary significantly, so the domain embedding must capture detailed object-specific information.

For most DA tasks (e.g., BDD day-to-night, Sim to Cityscapes, Cityscapes to Foggy Cityscapes), the domain differences are often driven by a single factor, such as time or weather, and the objects within the same class tend to have more consistent features. In these cases, the domain embeddings extracted by the vision encoder may be too detailed and could potentially conflict with the input layout. Therefore, our method may not be optimal for these tasks. It could be further improved by using more asymmetric training to remove redundant information, or by using learnable embeddings to represent different domains directly.

[1] Bdd100k: A diverse driving dataset for heterogeneous multitask learning. CVPR2020

[2] 2pcnet: Two-phase consistency training for day-to-night unsupervised domain adaptive object detection. CVPR2023

[3] ISP-Teacher: Image Signal Process with Disentanglement Regularization for Unsupervised Domain Adaptive Dark Object Detection. AAAI2024

[4] Cross domain object detection by target-perceived dual branch distillation. CVPR2022

[5] Unbiased mean teacher for cross-domain object detection. CVPR2021

[6] Cross-domain adaptive teacher for object detection. CVPR2022

---

### Comment · Area_Chair_nMMa · 2024-11-24
**discussion**

Dear reviewers,

Thank you for your contribution. Soon the discussion period is about to end. Please go over the response from the authors and initiate discussion.

regards

AC

---

### Meta-Review · Area_Chair_nMMa · 2024-12-22

**Metareview:**

Dear authors,

Thank you for submitting the draft. The draft received 5, 3, 6, 6. Therefore, average reviewers' rankings and comments by the reviewers indicate that the draft is not ready for publication at this stage. Two of the reviewers did not took part in the discussion, however, other two reviewers also only assigned 6 and 5 (marginally above and marginally below acceptance level).
We encourage authors to revise the draft, we hope comments by reviewers will help improve it.

regards

AC

**Additional Comments On Reviewer Discussion:**

Hi,

 r2nh believed that reviewer's comments went unaddressed and hold's "hold a negative stance towards the acceptance of this paper."
XiWb increased the score to 6 (marginally above the acceptance level).
Two reviewers did not took part in discussion, one assigned 6 other one 3.
Initially main concern appears to be lack of suffcient experiments, authors shared other experimental results in the rebuttal. However, were not able to convince the reviewers enough to give them high rating.

regards
AC

---

### Decision · Program_Chairs · 2025-01-22

Reject